# Provably Safe Generative Sampling with Constricting Barrier Functions

## Abstract

Flow-based generative models, such as diffusion models and flow matching models, have achieved remarkable success in learning complex data distributions. However, a critical gap remains for their deployment in safety-critical domains: the lack of formal guarantees that generated samples will satisfy hard constraints. We propose a safety filtering framework that acts as an online shield for any pre-trained generative model. Our key insight is to co-operate with the generative process rather than override it. We define a constricting safety tube that is relaxed at the initial noise distribution and progressively tightens to the target safe set at the final data distribution, mirroring the coarse-to-fine structure of the generative process itself. By characterizing this tube via Control Barrier Functions (CBFs), we synthesize a feedback control input through a convex Quadratic Program (QP) at each sampling step. As the tube is loosest when noise is high and intervention is cheapest in terms of control energy, most constraint enforcement occurs when it least disrupts the model's learned structure. We prove that this mechanism guarantees safe sampling in discrete-time. The minimum-norm control synthesized at each step minimizes the per-step contribution to the KL divergence between the safe and original distributions. Across all experiments, we observe 100% constraint satisfaction. Our framework applies to any pre-trained flow-based sampling scheme requiring no retraining or architectural modifications. We validate the approach across constrained image generation, physically-consistent trajectory sampling, and safe robotic manipulation policies, achieving 100% constraint satisfaction while preserving semantic fidelity.

## 1 Introduction

Flow-based generative models such as diffusion models (Ho et al., 2020; Song et al., 2021), flow matching (Lipman et al., 2023), and continuous normalizing flows (Papamakarios et al., 2021) have redefined the state-of-the-art in learning complex, high-dimensional distributions. By transforming a simple prior noise distribution into a structured data distribution through a sequence of infinitesimal steps, these models excel in tasks such as molecular design (Weiss et al., 2023), high-fidelity image synthesis (Ho et al., 2022), and control policies for robots (Chi et al., 2025). The deployment of these models increasingly requires constraint satisfaction across diverse applications. In content generation, constraints ensure human alignment, which requires filtering harmful content in images (Schramowski et al., 2023) or preventing toxic text generation (Li et al., 2025b). In safety-critical physical systems such as robotics (Janner et al., 2022; Chi et al., 2025) or autonomous navigation (Liao et al., 2025), constraints encode inviolable physical laws or safety specifications such as dynamical feasibility, joint limits, collision avoidance, smoothness, etc.

Traditional soft guidance techniques, such as classifier-based, classifier-free (Dhariwal & Nichol, 2021; Ho & Salimans, 2021) or reward-weighted guidance (Yuan et al., 2023), merely act as probabilistic incentives. While they bias the model toward desired regions, they cannot provide provable guarantees of feasibility. Conversely, projection-based methods can guarantee safety by formally projecting completed samples onto a safe manifold (Fishman et al., 2023; Utkarsh et al., 2025), yet they often suffer from significant computational overhead and introduce large distributional shifts.

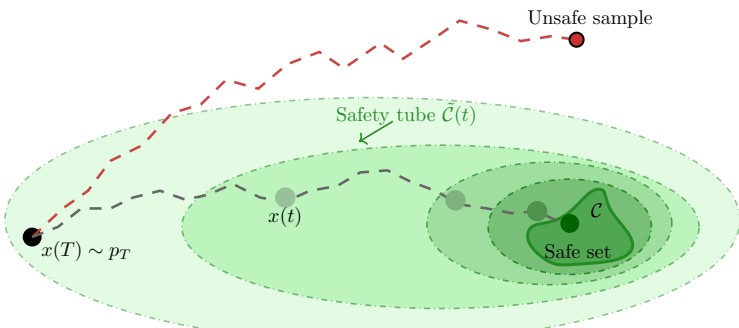

Figure 1: *Constrained generative sampling with a constricting safety tube.* The flow-based sampling process transforms noise sample $x(T)$ into data $x(0)$. *Unconstrained sampling process* can produce unsafe samples that violate constraints, landing outside the required safe set. Our control barrier function-guided sampling augments the generative dynamics with feedback control inputs $u$ that maintain the sample within the constricting safety tube $\tilde{\mathcal{C}}(t)$ (green region) throughout the sampling process, guaranteeing safe samples and minimal perturbation of the learned sampling process.

To address this gap, we propose a guidance scheme motivated by safety filtering in control theory. By leveraging Control Barrier Functions (CBFs), we synthesize a feedback control input through a Quadratic Program (QP) that aims to retain the fidelity of the original model. Our key insight is the use of a constricting safety tube $\tilde{\mathcal{C}}(t)$ that is relaxed at the high-noise regime ($t = T$) and progressively constricts to the target safe set $\mathcal{C}$ at the data distribution ($t = 0$). As illustrated in Figure 1, this ensures that the final sample belongs to the safe set while formally bounding the distributional shift between the original model and the safe distribution. We frame the guidance problem as a problem of control synthesis. The goal is to inject a feedback control input $u$ into the sampling process to render the safety tube $\tilde{\mathcal{C}}(t)$ invariant, ensuring that trajectories provably remain within the safety tube for the entire duration. CBFs provide the mathematical framework needed to synthesize such control inputs with formal safety certificates regarding the membership of $x(t)$ in $\tilde{\mathcal{C}}(t)$. Our framework ensures that the sample at the final sampling step $x(0)$ lies within the safe set $\mathcal{C}$ which satisfies the required constraints. At each sampling step, we solve a constrained optimal control problem that minimizes control effort to preserve distributional fidelity while enforcing a CBF constraint that guarantees $x(t) \in \tilde{\mathcal{C}}(t)$ for all $t \in [0, T]$.

Unlike prior CBF-based approaches (Xiao et al., 2023; Yang et al., 2025; Dai et al., 2025) that employ prescribed-time convergence or post-hoc trajectory correction, our method enforces strict invariance of a constricting safety tube that tightens during the sampling process, providing safety guarantees on sampling. Our main contributions are:

1. *Provably safe sampling:* For any closed $\mathcal{C}$, we prove that our guidance mechanism based on CBFs ensures that the final sample $x(0) \in \mathcal{C}$ exactly, for every realized noise sequence in discrete time (and deterministically when $g \equiv 0$). The continuous-time formulation provides design intuition through a formal invariance result. Importantly, we make no assumptions about the convexity of the safe set.

2. *Cooperation with the generative process:* Our constricting safety tube mirrors the coarse-to-fine structure of flow-based sampling, concentrating constraint enforcement in the high-noise regime where interventions are distributionally cheap. This ensures that the model retains full authority over the semantic structure and fine details. We prove that our minimum-norm control minimizes the per-step contribution to the KL divergence bound between the safe and original distributions.

3. *Modular guidance*: Our guidance scheme can be applied to any pre-trained flow-based generative model at sampling time, requiring no retraining or architectural modifications.

We validate our approach on three different experiments: simulation of nonlinear physics, constrained image generation, safety-critical planning in robotics.

## 2 Related Work

The literature on enforcing constraints in generative models is rapidly growing, typically bifurcated into methods that provide probabilistic incentives and those that formally enforce hard constraints.

**Soft guidance and probabilistic steering.** Soft guidance methods generally modify the sampling process by adding a penalty or a steering term that biases the model toward desired regions of the data distribution. Classifier guidance (Dhariwal & Nichol, 2021) utilize the probabilistic confidence from a classifier, or implicitly compute confidence in classifier-free guidance Ho & Salimans (2021) to steer the score function towards specific attributes or labels. Similarly, reward-guided methods Yuan et al. (2023) prioritize trajectories that maximize rewards, while gradient-based approaches (Guo et al., 2024) steer sampling via optimization objectives. While these methods effectively increase the likelihood of constraint satisfaction, they do not provide formal guarantees of admissibility. They lack a mechanism to explicitly ensure safe sampling. They are fundamentally probabilistic and may still produce samples that violate safety guidelines, making them unsuitable for safety-critical hardware where failure is impermissible. A growing body of inference-time alignment methods steers sampling toward high-reward regions without retraining Uehara et al. (2025). These include value-based and derivative-free reward guidance (Li et al., 2024), which bias the trajectory toward a reward without altering the model. A recurring theme in this line of work is the tension between reward optimization and generation quality or diversity: pushing hard toward a reward degrades sample quality, while staying close to the prior under-optimizes the objective (Kim et al., 2025). This tradeoff motivates our quantitative study of how hard guidance affects the generated distribution (Section 5.2.3), where we report FID, KID, and Vendi scores before and after guidance. Crucially, all of these reward-based methods remain probabilistic incentives: they raise the likelihood of constraint satisfaction but offer no admissibility certificate.

**CBF-based guidance in sampled robotic planning and control.** A recent line of literature in sampling based robotic planning and control have increasingly turned to CBFs (Mizuta & Leung, 2024; Dai et al., 2025; Xiao et al., 2023; Yang et al., 2025) to transform soft heuristics into formal safety certificates. Mizuta & Leung (2024) use a CBF-CLF framework to define a reward function, which is in turn used for reward-based guidance. Both Dai et al. (2025) and Xiao et al. (2023) employ prescribed-time CBF mechanisms with singular class-$\mathcal{K}$ functions that enforce convergence to the safe set by a deadline, requiring high-gain feedback that can lead to aggressive steering. Yang et al. (2025) adopts a prediction-correction architecture that generates a candidate path in latent space and then corrects it in a separate, post-hoc phase. In contrast, our method enforces strict invariance throughout the entire sampling process by utilizing a constricting safety tube. This ensures the particle remains safe at every integration step, rather than only at the final time. By using a minimum-norm control synthesis objective, we maintain maximal model fidelity without the need for high-gain steering. Our approach integrates the safety barrier directly into the sampling dynamics as a control intervention. We prove that our method minimizes the per-step contribution to the KL divergence bound between the safe and learned distributions. A concurrent control-theoretic approach is HardFlow (Li et al., 2025a), which casts hard-constrained sampling as a trajectory-optimization problem and uses a single-step model-predictive-control surrogate to enforce constraints at the terminal time. We differ in two respects: we render a constricting tube invariant at *every* step rather than only terminally, and our per-step intervention is a minimum-norm quadratic program with a closed-form solution (Remark 2) rather than an optimal-control surrogate whose guarantee holds up to a separate approximation error.

# 3 Preliminaries: Flow-based sampling and CBFs

Here we introduce essential mathematical preliminaries. We introduce generative sampling from the perspective of dynamical systems and the use of CBFs for safe control.

## 3.1 Flow-based sampling as a dynamical system

We formalize generative sampling as the trajectory of an autonomous system that transforms a prior noise distribution into a target data distribution. We define $t \in [0, T]$ as the sampling time, where $p(T)$ is the initial noise distribution at $t = T$, and $p(0)$ is the final data distribution at $t = 0$. The sampling process is governed by a stochastic differential equation (SDE):

$$\mathrm{d}x = f_\theta(x, t)\mathrm{d}t + g(t)\mathrm{d}w, \tag{1}$$

where the sample is $x(t) \in \mathbb{R}^n$, and $f_\theta(x, t)$ is the drift vector field. Further, $\mathrm{d}w$ is a reverse-time Wiener process (Song et al., 2021), reflecting that $t$ decreases from $T$ to 0 during sampling. The noise schedule $g(t)$ depends on $t$ alone and is independent of the state $x$, so the noise enters the dynamics additively. This state-independence will play a central role in our analysis: it ensures that the noise contribution to the trajectory is decoupled from the state, a property we exploit both in the continuous-time and, more importantly, in the rigorous discrete-time analysis. We are able to capture a large family of flow-based generative models using this notation. For example, in score-based diffusion models (Song et al., 2021), $f_\theta(x, t) = -\frac{1}{2}x - g^2(t)\nabla_x \log p(x, t)$ where the score $\nabla_x \log p(x, t)$ is approximated by a neural network. In flow matching (Lipman et al., 2023), $f_\theta = \mathrm{NN}_\theta(x, t)$ directly learns the velocity field that generates optimal transport paths between noise and data distributions. For deterministic frameworks like flow matching, the noise term vanishes, i.e. $g(t) = 0$, reducing the dynamics to an ordinary differential equation (ODE).

While the SDE (1) describes individual sample paths, it induces a corresponding evolution of the population's probability density $p(x, t)$ as $t$ decreases from $T$ to 0. This progression has an important structural property: in the high-noise regime, the broad support of $p(x, t)$ means the model establishes only global, coarse structure. As $t \to 0$ and the distribution concentrates toward the data distribution $p(0)$, the model progressively resolves finer details. We refer to this as the *coarse-to-fine* structure of flow-based sampling, and our constricting safety tube in section 4 is designed to mirror this progression. This coarse-to-fine progression has a direct consequence for constrained sampling: interventions applied during the high-noise regime, where the model has not yet committed to fine-grained structure, are less disruptive than those applied near $t = 0$ when the sample has nearly converged.

In practice, samples are generated by numerically simulating (1) in discrete-time. We discretize the interval $[0, T]$ into $K$ steps of size $\Delta t = T/K$, with time indices $t_k = k\Delta t$ for $k = K, K-1, \ldots, 0$, and use the notation $x_k = x(t_k)$. The Euler–Maruyama scheme for (1) gives the update

$$x_{k-1} = x_k - f_\theta(x_k, t_k)\,\Delta t + g(t_k)\sqrt{\Delta t}\,\xi_k, \qquad \xi_k \sim \mathcal{N}(0, I), \tag{2}$$

Here $k$ decreases in the sampling process, hence the minus sign in the drift. In the simulated sampler, the noise increment $\xi_k$ is drawn from a standard Gaussian distribution.

## 3.2 Control Barrier Functions for Set Invariance

Here we review the theory of control barrier functions to design feedback controllers to enforce reverse invariance of the constraint set. The design approach we develop in the sequel modifies and extends the standard control barrier function (CBF) framework, which we review here for completeness. Consider a system whose state $x(t) \in \mathbb{R}^n$ evolves as $t$ decreases from $T$ to 0, governed by:

$$\mathrm{d}x = f(x, u)\,\mathrm{d}t, \tag{3}$$

where $u \in \mathbb{R}^m$ is the control input. Since $t$ decreases during sampling, $\mathrm{d}t < 0$ throughout, and $f(x, u)$ represents the drift of the reverse-time dynamics, consistent with the sampling process (1). Assume that we

are given a set $\mathcal{C} \subset \mathbb{R}^n$ of "safe" states, given by the superlevel set of a continuously differentiable function $h : \mathbb{R}^n \to \mathbb{R}$:

$$\mathcal{C} = \{x \in \mathbb{R}^n \mid h(x) \geq 0\}. \tag{4}$$

We seek a feedback controller that keeps the state in $\mathcal{C}$ throughout sampling, designed via control barrier functions Ames et al. (2016; 2019). We say $\mathcal{C}$ is *reverse invariant*[1] if, for any terminal condition $x(T) \in \mathcal{C}$, the trajectory satisfies $x(t) \in \mathcal{C}$ for all $t \in [0, T]$ as $t$ decreases from $T$ to 0; equivalently, $h(x(T)) \geq 0$ implies $h(x(t)) \geq 0$ for all $t \in [0, T]$.

We say $h$ is a *reverse-time control barrier function*[2] if for all $x \in \mathcal{C}$ there exists $u \in \mathbb{R}^m$ such that

$$\nabla h(x) \cdot f(x, u) \leq \gamma(h(x)), \tag{5}$$

where $\gamma$ is a class-$\mathcal{K}$ function. Any feedback controller satisfying (5) renders $\mathcal{C}$ reverse invariant. Intuitively, on the boundary $\partial\mathcal{C}$ where $h(x) = 0$ the condition forces $\mathrm{d}h/\mathrm{d}t \leq 0$, which in reverse time (decreasing $t$) keeps $h$ non-decreasing, so the trajectory cannot leave $\mathcal{C}$; in the interior $\gamma(h(x)) > 0$ permits $h$ to vary while continuity of $\gamma$ drives $\mathrm{d}h/\mathrm{d}t \to 0$ as $x \to \partial\mathcal{C}$.

The CBF condition (5) also yields guarantees for discretizations of (3). For a system simulated on a discrete time grid $t_k = k\Delta t$ with state update $x_{k-1} = x_k - f(x_k, u_k)\Delta t$ (reverse-time convention $\mathrm{d}t < 0$), and a linear class-$\mathcal{K}$ function $\gamma(h) = \alpha h$ with $\alpha > 0$, the corresponding discrete-time CBF condition is

$$h(x_{k-1}) \geq (1 - \alpha\,\Delta t)\,h(x_k). \tag{6}$$

This ensures $h(x_k) \geq 0$ implies $h(x_{k-1}) \geq 0$ for all $\alpha\Delta t \leq 1$, the discrete analogue of reverse invariance, and reduces to (5) as $\Delta t \to 0$. Condition (6) has two advantages in our setting. First, $h(x_{k-1})$ is determined by the simulated dynamics including the observed noise increment, so the condition translates into an algebraic constraint on $u_k$ once $\xi_k$ is drawn. Second, invariance follows by induction on $k$ rather than via continuous-time theorems requiring a differentiable trajectory. We use (6) as the basis for our rigorous safety analysis, while (5) provides the design intuition.

## 4 Guidance of flow-based models via constricting CBFs

Having established the sampling dynamics (1) and the CBF framework for safe control, we address the central question: how can we enforce hard constraints on the output of a pre-trained generative model without modifying the model itself? The key observation is that by introducing a control input to the sampling SDE (1), we arrive at a control-affine structure analogous to the control systems considered in Section 3.2. This additive control input $u$ can be designed to steer sampling trajectories toward the safe set while preserving the learned distribution. This is a control synthesis problem with the following guided sampling process:

$$\mathrm{d}x = [f_\theta(x, t) + u(x, \xi, t)]\mathrm{d}t + g(t)\mathrm{d}w, \quad t \in [0, T], \tag{7}$$

where $f_\theta$ is the unconstrained drift learned by the generative model, $g(t)$ is the noise schedule, $w$ is a Wiener process, and $u(x, \xi, t)$ is the feedback control that depends on the current state $x$ and the realized noise[3] $\xi$. The challenge is to design $u$ so that it provides formal safety guarantees while remaining minimally invasive, intervening only when necessary and as little as possible.

Note that our proposed feedback controller depends both on the state of the system and the noise process. Although this assumption may be restrictive in settings where the feedback controller regulates a physical

---

[1]Standard literature in safety-critical control treats systems evolving forward in time, deriving conditions for forward invariance. Since generative sampling evolves backward in time, we instead enforce reverse invariance. Our presentation follows Ames et al. (2019), with the CBF condition (5) modified for the reversed time direction.

[2]For brevity we henceforth refer to reverse-time control barrier functions simply as CBFs, since we only deal with systems evolving backward in time.

[3]Here $\xi(t)$ is the Gaussian white-noise process whose integral recovers the driving Wiener process, $\int_T^t \xi(s)\,\mathrm{d}s = w(t)$. This is the formal identification of white noise with the increments of Brownian motion.

system, and the noise cannot be directly measured, in our application we have full knowledge of the noise process since the system is being simulated on a computer. This observe-then-act structure is what enables the rigorous discrete-time safety guarantees we establish in the next section. Controllers depending only on the state cannot in general provide analogous guarantees (cf. Remark 1).

## 4.1 Constricting CBFs for flow-based generative sampling

A fundamental challenge in generative sampling is that the initial prior noise $x(T)$ is drawn from a distribution such as the standard Gaussian. This means that $x(T)$ almost certainly resides outside the target safe set $\mathcal{C}$. Traditional CBF formulations for static sets would require an immediate, high-energy intervention to push $x(t)$ close to $\mathcal{C}$, which would violate the model's generative intent and produce samples that no longer represent the learned distribution. Our framework is motivated by the need for minimal interventions and a smooth approach towards $\mathcal{C}$. We achieve this by defining a constricting barrier $\tilde{h}(x,t) = h(x) + \epsilon(x(T),t)$. This is a time-varying barrier that constricts with the sampling process. We characterize safety through a constricting superlevel tube $\tilde{\mathcal{C}}(t)$ with respect to the constricting barrier $\tilde{h}$ as,

$$\tilde{\mathcal{C}}(t) = \{x \in \mathbb{R}^n \mid \tilde{h}(x,t) \geq 0\}. \tag{8}$$

We design $\tilde{\mathcal{C}}(t)$ such that initially, the set $\tilde{\mathcal{C}}(T)$ is sufficiently relaxed to contain the noise sample $x(T)$, and at the end of sampling, it recovers the target set $\mathcal{C}$, i.e., $\tilde{\mathcal{C}}(0) = \mathcal{C}$. Since the sampling process evolves in reverse time from $t = T$ to $0$, the safety tube $\tilde{\mathcal{C}}(t)$ must remain occupied by $x(t)$ as $t$ decreases, a property we term *reverse invariance*. As derived in Section 3.2, the reverse-time CBF condition requires bounding $\nabla \tilde{h} \cdot f$ from above. We design the constricting barrier function through a time-varying control barrier function.

**Definition 1 (Constricting barrier function)** *Given a CBF $h : \mathbb{R}^n \to \mathbb{R}$ and a terminal condition $x(T)$, a constricting barrier function is $\tilde{h}(x,t) = h(x) + \epsilon(x(T),t)$, where $\epsilon$ is a $C^1$ function satisfying:*

1. ***Initial feasibility:*** *$\epsilon(x(T),T) \geq \max(0, -h(x(T)))$.*

2. ***Recovery of target set:*** *$\epsilon(x(T),0) = 0$.*

3. ***Monotone constriction:*** *$\partial\epsilon/\partial t \geq 0$ for all $t \in [0,T]$. Since $t$ decreases from $T$ to $0$ during sampling, this means $\epsilon$ decreases as sampling progresses, constricting $\tilde{\mathcal{C}}(t)$ toward $\mathcal{C}$.*

*The associated constricting safety tube is $\tilde{\mathcal{C}}(t) = \{x \in \mathbb{R}^n \mid \tilde{h}(x,t) \geq 0\}$.*

The concept of a time-varying barrier that tightens onto the target set was introduced in the deterministic setting for prescribed-time safe control, where the constraint set is deformed on a fixed schedule so that safety is achieved by a deadline (Gadginmath et al., 2026). For flow-based generative sampling, the schedule is the sampling horizon itself. The following result, which is the reverse-time analog of (Gadginmath et al., 2026, Theorem 1), characterizes the continuous-time condition under which the sampling process (7) can be made invariant to the safety tube $\tilde{\mathcal{C}}(t)$ in the deterministic setting.

**Proposition 4.1 (Reverse invariance, deterministic case)** *Let $\tilde{h}$ be a constricting barrier function, $\alpha > 0$, and suppose $g \equiv 0$, so that the guided sampling process (7) reduces to the deterministic dynamics $\dot{x} = f_\theta(x,t) + u(x,t)$. If the control $u(x,t)$ satisfies*

$$\nabla_x \tilde{h}(x,t) \cdot (f_\theta(x,t) + u(x,t)) + \frac{\partial \tilde{h}(x,t)}{\partial t} \leq \alpha \tilde{h}(x,t), \tag{9}$$

*for all $(x,t)$, then the trajectory satisfies $x(t) \in \tilde{\mathcal{C}}(t)$ for all $t \in [0,T]$, and in particular $x(0) \in \mathcal{C}$.*

Proposition 4.1 establishes the design rigorously in the deterministic setting. In the stochastic case ($g > 0$), the same construction motivates enforcing the analogous condition along each realized noise path,

$$\nabla_x \tilde{h}(x,t) \cdot (f_\theta(x,t) + u(x,\xi,t) + g(t)\xi(t)) + \frac{\partial \tilde{h}(x,t)}{\partial t} \leq \alpha \tilde{h}(x,t), \tag{10}$$

where $g(t)$ is the noise schedule and $\xi(t)$ denotes the realized noise (c.f. footnote 3) along the trajectory, which the controller observes before acting. A full formalization of this continuous-time stochastic setting requires interpreting the trajectory as the solution of a constrained SDE, for which the Skorokhod problem (Tanaka, 1979) provides the appropriate framework. We instead establish the rigorous results in Section 4.2 for the discretization, the setting relevant to samplers used in practice.

Proposition 4.1 and the stochastic condition (10) motivate the design: any control $u$ satisfying the CBF condition (10) would deliver $x(0) \in \mathcal{C}$, regardless of the convexity of $\mathcal{C}$, the architecture of $f_\theta$, or the location of the initial noise sample $x(T)$. The key mechanism is the interplay between the constriction and the gain $\alpha$. At the onset of sampling ($t \approx T$), $\tilde{h} = h + \epsilon$ is large due to the relaxation $\epsilon$. As sampling progresses, $\epsilon \to 0$ and $\tilde{h} \to h$, so $\alpha\tilde{h}$ shrinks and the constraint tightens toward the standard CBF condition for $\mathcal{C}$ guided by the feedback control $u(x, \xi, t)$. By this point, the learned model $f_\theta$ has already resolved the trajectory close to $\mathcal{C}$, and the control $u$ needs only to enforce the final constraint boundary. Many fixed-time convergence schemes rely on singular class-$\mathcal{K}$ functions that blow up as $t \to 0$ (Xiao et al., 2023; Dai et al., 2025). In contrast, our $C^1$ constriction mimics the coarse-to-fine generation scheme of typical flow-based models, ensuring that any feasible control remains bounded and minimally invasive. We emphasize that the safe sampling guarantee concerns membership in the safe set $\mathcal{C} = \{x : h(x) \geq 0\}$ defined in the space in which the sampling process evolves, when the barrier $h$ is expressed directly on the model's output (data) space. This co-location is essential: the constricting CBF condition (10) constrains the barrier value, and the control steers along $\nabla_x \tilde{h}$ in the same space the sampler integrates. The certificate is therefore exact precisely when the barrier and the state share a space. In our image experiments this space is pixel space, in the Lorenz experiment the trajectory space, and in the robotics experiment the action space, and in each the constraint is imposed directly on the sampled variable, so the guarantee holds exactly. A distinct case arises when the safe set is defined in one space while the sampler evolves in another. The most important instance is the latent diffusion model, where the sampler evolves a latent $z$ but the constraint $\mathcal{C}$ lives in the decoded image space. The barrier the sampler sees is then the composition $h \circ D$, and its gradient $\nabla_z(h \circ D)$ is the decoder-pulled-back image-space gradient. The discrete-time guarantee certifies membership in the latent preimage $D^{-1}(\mathcal{C})$, which coincides with $\mathcal{C}$ exactly when the decoder $D$ is invertible. For the VAE decoders used in latent diffusion, $D$ is not invertible, and the certificate in latent space transfers to the decoded image up to the decoder's reconstruction error. We discuss this in Section 6 and demonstrate it empirically.

**Remark 1 (Comparison to other approaches that extend CBFs to stochastic systems)** *Unlike works that consider designing feedback controllers that enforce safety of stochastic dynamical systems (Clark, 2021), our approach assumes that the feedback controller has knowledge of the state as well as the noise. The latter is reflective of the operational reality of a sampling algorithm, where unlike the case of a physical system subject to disturbances, the noise is simulated on a computer and one is capable of observing it. The use of noise-dependent feedback controllers allows us to obtain rigorous guarantees of invariance of the constricting tube in discrete-time, unlike controllers depending only on the state which cannot enforce invariance exactly or almost-surely (So et al., 2023).*

The stochastic condition (10) reflects the design principle that, at each instant, the controller observes the noise and acts on the observed value, rather than acting only in expectation over future noise. While Gaussian noise has unbounded support and extreme realizations could theoretically require large control effort, our experiments (Section 5) demonstrate reliable feasibility across diverse applications. Alternative formulations that use model predictive control require stochastic CBFs (Prajna et al., 2004) to predict the behavior of model and the noise. Here, chance constraints can provide formal high-probability bounds by incorporating safety margins proportional to the noising scheme $g(t)$. We defer such extensions to future work.

We now discuss candidates for the constriction scheme $\epsilon(x(T), t)$. Let $\epsilon_0 = \max(0, -h(x(T)))$ denote the initial constraint violation. A key practical advantage of Definition 1 is that its conditions are mild enough to be satisfied by a broad family of simple, closed-form functions. We present three such candidates: (1) *Linear*: $\epsilon(x(T), t) = \epsilon_0 \cdot (t/T)$ with constant constriction rate, (2) *Exponential*: $\epsilon(x(T), t) = \epsilon_0 \cdot (e^{\lambda t/T} - 1)/(e^\lambda - 1)$ for $\lambda > 0$, which front-loads relaxation when $\lambda > 1$, providing aggressive early intervention and gentler later refinement, (3) *Polynomial*: $\epsilon(x(T), t) = \epsilon_0 \cdot (t/T)^p$ for $p \geq 1$, which back-loads constriction when $p > 1$, offering slower initial relaxation and faster final convergence. All schemes satisfy Definition 1 by construction,

and the shape parameter ($\lambda$ or $p$) gives practitioners direct control over when along the sampling trajectory the constriction pressure is concentrated.

To preserve the fidelity of the pre-trained model, we implement the *minimum-norm* control that satisfies the CBF condition (10). Intervening as little as possible at each step is what lets the framework cooperate with the generative process rather than override it. Flow-based models perform coarse-to-fine refinement: the high-noise phase establishes global structure, while the low-noise phase resolves fine detail. The constricting tube mirrors this progression by design, maximally relaxed when noise is high and tightening only as the model refines local detail. Because the tube is loose in the high-noise regime, the CBF condition can be met with a small intervention early on, when the sample has not yet committed to fine structure; by the time the model resolves fine detail, the tube has already guided the trajectory close to the safe set and little to no intervention remains. The model thus retains full authority over the structure and detail that determine sample quality. This stands in contrast to projection-based methods, which apply corrections independently of the noise schedule and therefore pay the same cost at every step, disrupting global structure when applied early and overriding fine detail when applied late.

## 4.2 Constricted sampling in discrete-time

We now establish the rigorous safety and distribution-shift guarantees of our framework at the level of the simulated Euler–Maruyama process (2). The simulated sampler observes each noise realization $\xi_k$ before synthesizing the control $u_k$. This observe-then-act structure enables pathwise guarantees for safe sampling. The time discretization is $t_k = k\Delta t$ for $k = 0, 1, \ldots, K$ with $\Delta t = T/K$, and the guided update is,

$$x_{k-1} = x_k - [f_\theta(x_k, t_k) + u_k]\Delta t + g(t_k)\sqrt{\Delta t}\,\xi_k, \qquad \xi_k \sim \mathcal{N}(0, I), \tag{11}$$

where the control $u_k = u_k(x_k, \xi_k, t_k)$ is a measurable function of the observed state and noise. Let $\tilde{h}_k := \tilde{h}(x_k, t_k) = h(x_k) + \epsilon(x_K, t_k)$ denote the value of the constricting barrier at step $k$. The *discrete-time constricting CBF condition* on the controlled update is

$$\tilde{h}(x_{k-1}, t_{k-1}) \geq (1 - \alpha\Delta t)\,\tilde{h}_k, \tag{12}$$

where $\alpha > 0$. This is the exact discrete-time analogue of the continuous-time CBF condition (5): it requires the next-step barrier value to be at least a $(1 - \alpha\Delta t)$ fraction of the current value, so that $\tilde{h}$ can decrease no faster than at rate $\alpha$ per unit time.

**Theorem 4.2 (Discrete-time reverse invariance)** *Suppose at every step $k = K, K-1, \ldots, 1$, the control $u_k(x_k, \xi_k, t_k)$ is chosen such that condition (12) holds, with $\alpha\Delta t \leq 1$. Then, $x_k \in \tilde{\mathcal{C}}(t_k)$ for every $k$, and in particular $x_0 \in \mathcal{C}$.*

**Proof.** The initial feasibility condition of Definition 1 gives $\tilde{h}_K = h(x_K) + \epsilon(x_K, T) \geq 0$, so $x_K \in \tilde{\mathcal{C}}(t_K)$. For the inductive step, assume $\tilde{h}_k \geq 0$. Condition (12) together with $\alpha\Delta t \leq 1$ gives

$$\tilde{h}(x_{k-1}, t_{k-1}) \geq (1 - \alpha\Delta t)\,\tilde{h}_k \geq 0,$$

so $x_{k-1} \in \tilde{\mathcal{C}}(t_{k-1})$. By induction, $x_k \in \tilde{\mathcal{C}}(t_k)$ for all $k$. At $k = 0$, the target-recovery condition $\epsilon(x_K, 0) = 0$ gives $\tilde{h}_0 = h(x_0) \geq 0$, hence $x_0 \in \mathcal{C}$. The argument applies pathwise to every realization $\{\xi_k\}_{k=1}^K$, and the conclusion follows for every such realization. ∎

We now characterize the distributional shift induced by guidance.

**Theorem 4.3 (Discrete-time distribution shift)** *Let $p(0)$ and $p_{\text{safe}}(0)$ denote the terminal marginal distributions of $x_0$ under the unconstrained sampling process (2) and the guided process (11), respectively. Then*

$$D_{\text{KL}}\big(p_{\text{safe}}(0) \,\|\, p(0)\big) \leq \frac{1}{2}\,\mathbb{E}\left[\sum_{k=1}^K \frac{\|u_k\|^2}{g(t_k)^2}\,\Delta t\right], \tag{13}$$

*where the expectation is taken over the guided sampling path.*

**Proof.** Let $P_K$ and $Q_K$ denote the joint distributions of the unconstrained and guided sampling paths $(x_K, x_{K-1}, \ldots, x_0)$, generated by (2) and (11) respectively. We first bound the path-measure divergence $D_{\mathrm{KL}}(Q_K \| P_K)$ and then contract to the terminal marginals. Both path measures are determined by the joint law of the noise sequence $\{\xi_k\}_{k=1}^K$ and the initial state $x_K$, where $x_K \sim p(T)$ and $\xi_k \sim \mathcal{N}(0, I)$ are independent. Condition on $(x_k, \xi_k)$ at each step. Under this conditioning, $u_k = u_k(x_k, \xi_k, t_k)$ is a deterministic vector, and the controlled transition reduces to a deterministic shift of the uncontrolled transition: $x_{k-1}$ under $Q_K$ equals $x_{k-1}$ under $P_K$ minus $u_k \Delta t$. Viewing this conditionally as a comparison of two Gaussians with equal covariance $\Sigma = g(t_k)^2 \Delta t \cdot I$ and mean shift $u_k \Delta t$,

$$D_{\mathrm{KL}}\big(Q_k(\cdot | x_k, \xi_k) \, \| \, P_k(\cdot | x_k, \xi_k)\big) = \frac{1}{2} \frac{\|u_k\|^2}{g(t_k)^2} \Delta t.$$

Equal covariance holds because the control modifies only the drift, leaving the noise schedule $g(t_k)$ unchanged; this is the structural property responsible for the $1/g(t_k)^2$ cost factor. The transitions share support, so the divergence is finite. Taking expectation over $(x_k, \xi_k)$ under $Q_K$ and summing via the chain rule for KL divergence applied to the joint path measure (Cover & Thomas, 2006, Theorem 2.5.3),

$$D_{\mathrm{KL}}(Q_K \| P_K) = \frac{1}{2} \mathbb{E}_{Q_K} \left[ \sum_{k=1}^K \frac{\|u_k\|^2}{g(t_k)^2} \Delta t \right].$$

Terminal marginals $p(0)$ and $p_{\mathrm{safe}}(0)$ are the pushforwards of $P_K$ and $Q_K$ under the projection $(x_K, \ldots, x_0) \mapsto x_0$, so by the data-processing inequality $D_{\mathrm{KL}}(p_{\mathrm{safe}}(0) \| p(0)) \leq D_{\mathrm{KL}}(Q_K \| P_K)$, yielding (13). ∎

Subject to the discrete CBF condition (12), the per-step minimizer of the integrand $\|u_k\|^2 / g(t_k)^2$ is the greedy minimizer of the bound (13) at each step. While this strategy does not guarantee global optimality over the entire sampling horizon, which would require an optimal control formulation accounting for future noise realizations, it yields the tightest single-step bound on the distributional shift. The factor $1/g(t_k)^2$ in (13) reveals that control interventions are cheapest in distributional terms when the noise level $g(t_k)$ is large. This is precisely the structure that the constricting tube is designed to exploit. Flow-based models perform coarse-to-fine refinement: the high-noise phase establishes global structure, while the low-noise phase resolves fine details. The constricting tube mirrors this progression, maximally relaxed when noise is high and tightening only as the model refines local details. As the tube is loosest precisely when intervention is cheapest, most constraint enforcement is absorbed during the high-noise regime at minimal distributional cost. By the time the model enters the low-noise regime where fine details are resolved, the tube has already guided the trajectory close to the safe set, and $u_k^* \approx 0$. The model thus retains full authority over the structure and details that determine sample quality. When the safe set $\mathcal{C}$ overlaps significantly with $\mathrm{supp}(p(0))$, the learned drift $f_\theta$ already steers samples toward safety, requiring minimal control effort throughout. When $\mathcal{C}$ is disjoint from $\mathrm{supp}(p(0))$, the QP identifies the closest safe distribution under the greedy minimization.

The bound (13) is stated for the stochastic case when $g > 0$. As $g(t_k) \to 0$ the per-step factor $1/g(t_k)^2$ diverges, and the bound degenerates. This is a property of the deterministic limit: the guided and unguided terminal laws are pushforwards of the same initial distribution under two distinct deterministic flows, and such pushforwards are in general mutually singular, so $D_{\mathrm{KL}}(p_{\mathrm{safe}}(0) \, \| \, p(0))$ is ill-defined and KL divergence is no longer the appropriate measure of distributional shift. The natural analogue is the integrated control energy $\int_0^T \|u(x, t)\|^2 \, dt$, a transport cost rather than a likelihood ratio. Minimizing $\|u_k\|^2$ pointwise, as the QP (15) does, then corresponds to the minimum-$L_2$ perturbation of the learned velocity field that keeps the trajectory within the tube: the smallest deviation from the flow-matching drift consistent with safety. The minimum-norm objective is therefore the right per-step criterion in both regimes. It minimizes the KL bound when $g > 0$ and the control energy when $g \equiv 0$. This lets our algorithm serve deterministic and stochastic samplers without modification.

Our refinement of the CBF over the sampling horizon stands in contrast to projection-based methods, which apply corrections independently of the noise schedule and therefore pay the same distributional cost at every step, disrupting global structure when applied early and overriding fine details when applied late. The experimental consequence is visible in Figure 4: projection-based enforcement (b) satisfies the constraint but destroys semantic coherence, whereas our CBF-guided sampling (a, c) preserves realistic scene structure by deferring to the model during the critical structure-forming phase.

### 4.3 Implementation: linearized QP and algorithm

Theorem 4.2 establishes that any control satisfying the exact discrete CBF condition (12) guarantees reverse invariance of the constricting tube. However, condition (12) is implicit in $u_k$: the next-step barrier value $\tilde{h}(x_{k-1}, t_{k-1})$ depends on $u_k$ through the nonlinear function $\tilde{h}$ applied to the updated state $x_{k-1}$ from (11). For general barrier functions $h$ this implicit constraint cannot be solved in closed form. We instead linearize the constraint in $u_k$ and synthesize the feedback controller as a convex quadratic program that enforces the linearized condition. A first-order Taylor expansion of $\tilde{h}$ around $(x_k, t_k)$ gives

$$\tilde{h}(x_{k-1}, t_{k-1}) \approx \tilde{h}_k + \nabla_x \tilde{h}_k \cdot (x_{k-1} - x_k) + \frac{\partial \tilde{h}}{\partial t}(x_k, t_k) \cdot (-\Delta t).$$

Substituting the Euler–Maruyama update (11) and the identity $\partial \tilde{h}/\partial t = \partial \epsilon/\partial t$, the exact discrete CBF condition (12) becomes the inequality, affine in $u_k$,

$$\tilde{h}_k + \nabla_x \tilde{h}_k \cdot \big( -[f_\theta(x_k, t_k) + u_k]\Delta t + g(t_k)\sqrt{\Delta t}\,\xi_k \big) - \frac{\partial \epsilon}{\partial t}(x_K, t_k)\Delta t \geq (1 - \alpha\Delta t)\,\tilde{h}_k. \tag{14}$$

The minimum-norm control synthesis is then the convex QP

$$\min_{u_k} \frac{1}{2}\|u_k\|^2 \quad \text{s.t.} \quad (14), \tag{15}$$

solved for the observed state–noise pair $(x_k, \xi_k)$ at each step $k$. This QP drives the algorithm at each sampling step. The complete procedure is given in Algorithm 1.

**Remark 2 (Feasibility and closed-form solution of the QP** (15)**)** *The linearized constraint* (14) *is a single linear inequality in $u_k$ of the form $a^\top u_k \leq b$, with $a = \nabla_x \tilde{h}_k\,\Delta t$ and $b$ collecting the remaining terms of* (14). *The QP* (15) *is therefore a minimum-norm problem with one linear constraint. When $b \geq 0$, the uncontrolled update already satisfies the linearized condition and $u_k^* = 0$. When $b < 0$, the QP admits the closed-form solution $u_k^* = (b/\|a\|^2)\,a$ whenever $a \neq 0$, which points along $-\nabla_x \tilde{h}_k$. Equivalently, $u_k^* = \min(0, b/\|a\|^2)\,a$. The regularity assumption $\nabla h(x) \neq 0$ for $x \in \partial\mathcal{C}$ ensures $a \neq 0$ near the constraint boundary where intervention is needed, and in the interior of $\tilde{\mathcal{C}}(t_k)$ where $\tilde{h}_k \gg 0$ the constraint is slack and $u_k^* = 0$. The QP is thus always feasible for any finite noise realization $\xi_k$. The closed-form solution makes the per-step cost negligible, and when the barrier decomposes across coordinates the QP separates into independent low-dimensional problems solved in parallel (Section 5.2.1).*

The linearized constraint (14) is a first-order Taylor approximation of the exact condition (12), and is therefore sufficient for the exact condition only up to the second-order remainder $\frac{1}{2}\Delta x^\top \nabla_x^2 \tilde{h}\,\Delta x$, where $\Delta x = x_{k-1} - x_k$ is the per-step increment. The order of this remainder differs between the two regimes: in the deterministic case ($g \equiv 0$), $\Delta x$ is of order $O(\Delta t)$ and the remainder is of order $O(\Delta t^2)$. In the stochastic case ($g > 0$), $\Delta x$ is of order $O(\sqrt{\Delta t})$ so the remainder is of order $O(\Delta t)$. In either regime the linearization error shrinks with $\Delta t$ and with the curvature $\nabla_x^2 \tilde{h}$. In practice, it is absorbed by the per-step margin from the gain term $\alpha\,\tilde{h}_k\,\Delta t$ together with the conservative initialization $\epsilon_0$. We do not establish a uniform bound on the accumulated violation over the horizon, as it depends on the learned vector field $f_\theta$. A rigorous discrete-time CBF analysis with explicit step-size-dependent bounds is left as a direction for future work. Empirically, across all experiments with step sizes ranging from $\Delta t = 0.01$ (Section 5.1) to $\Delta t = 0.005$ (Section 5.3), we observed zero constraint violations at the final sample, with the per-step linearization error reported in Table 1.

## 5 Experiments

We validate our framework across three domains that demonstrate its versatility and practical effectiveness. In each experiment, we apply our CBF-guided sampling to pre-trained, off-the-shelf generative models without any retraining or architectural modifications, demonstrating the modularity claimed in Contribution 2. For each experiment, our goal is to demonstrate that the safe sampling guarantees of Theorem 4.2 hold empirically across qualitatively distinct constraint types.

---

**Algorithm 1:** CBF-guided safe sampling

---

**Input:** Pre-trained model $f_\theta$, noise schedule $g(t)$, safe set $\mathcal{C} = \{x \mid h(x) \geq 0\}$, number of steps $K$, gain
$\qquad \alpha > 0$

**Output:** Safe sample $x_0 \in \mathcal{C}$

**1. Initialize:** Sample $x_K \sim p(T)$ (e.g., $x_K \sim \mathcal{N}(0, I)$), set $t_K = T$, $\Delta t = T/K$

**2. for** $k = K, K-1, \ldots, 1$ **do**

$\quad$ **3.** Sample noise: $\xi_k \sim \mathcal{N}(0, I)$

$\quad$ **4.** Compute noise-induced drift: $d_{\text{noise}} = g(t_k)\sqrt{\Delta t}\, \xi_k$

$\quad$ **5.** Compute constricting barrier: $\tilde{h}_k = h(x_k) + \epsilon(x_K, t_k)$

$\quad$ **6.** Solve safety-constrained QP:

$$\min_{u_k} \quad \frac{1}{2}\|u_k\|^2$$

$$\text{s.t.} \quad \tilde{h}_k + \nabla_x \tilde{h}_k \cdot (-[f_\theta(x_k, t_k) + u_k]\Delta t + d_{\text{noise}}) - \frac{\partial \epsilon}{\partial t}(x_K, t_k)\Delta t \geq (1 - \alpha\Delta t)\, \tilde{h}_k$$

$\quad$ **7.** Apply control and update state: $x_{k-1} = x_k - f_\theta(x_k, t_k)\Delta t - u_k^*\Delta t + d_{\text{noise}}$

$\quad$ **8.** Update time: $t_{k-1} = t_k - \Delta t$

**9. return** $x_0$

---

All experiments use the linear constriction $\epsilon(x(T), t) = \epsilon_0 \cdot \frac{t}{T}$ where $\epsilon_0 = \max(0, -h(x(T))) + c$, where $c > 0$ is a small positive margin. This margin prevents the barrier from operating exactly on its zero level set at initialization, where numerical precision could cause spurious violations. In all experiments, we set $c = 0.1$ and choose $\alpha = 0.5$. The QPs in Algorithm 1 are solved using CVXPY with the OSQP solver. Our implementation is available at [anonymized for review]. Our experiments were conducted on an Intel i9-9900 machine with 128GB RAM and an Nvidia Quadro RTX 4000 GPU.

## 5.1 Physics-consistent trajectory generation for the Lorenz system

In this experiment, we demonstrate the behavior of our guidance scheme on a synthetic testbed. We seek to generate trajectories of the Lorenz system with dynamics,

$$\begin{aligned}
\dot{z}_1 &= \sigma(z_2 - z_1) \\
\dot{z}_2 &= z_1(\rho - z_3) - z_2 \\
\dot{z}_3 &= z_1 z_2 - \beta z_3
\end{aligned} \tag{16}$$

where we set $\sigma = 10, \rho = 28, \beta = \frac{8}{3}$. This system exhibits a characteristic behavior called the Lorenz attractor, famously known as the butterfly effect, where it is highly sensitive to the initial condition. Our goal is to ensure that sampled trajectories satisfy the true physics encoded in (16). Given an initial condition $z(0) = [z_1(0)\ z_2(0)\ z_3(0)]^\top$, we seek to sample an entire trajectory up to 10 seconds of evolution. We use a physical time discretization $\Delta_l = 0.01$, which results in $L = 1000$ sampling steps. For the sake of brevity, we denote the state sampled at time $l\Delta_l$ as $z(l\Delta_l) = z^l$. We sample a full trajectory $x(0) = [z^{0\top}\ z^{1\top}\ \cdots\ z^{L\top}]^\top$. Note that $l$ is the sampling step in physical time for the Lorenz system, whereas we reserve $k$ for the sampling time for the diffusion model.

We select the Lorenz oscillator because its ground-truth solution is available via numerical integration, providing an exact reference for validating our guidance scheme. In practice, our framework targets settings where the governing equations are high-dimensional or computationally expensive to integrate, such as turbulent flows or multi-scale PDEs, and where a generative model serves as an efficient surrogate. Unlike traditional numerical methods that accumulate integration error over time, our approach generates the entire trajectory as a single high-dimensional sample. The Lorenz system thus serves as a controlled testbed to verify that our CBF-guided sampling can recover underlying physical laws and maintain temporal consistency across the entire sequence before deployment in more complex domains.

We train a diffusion model with synthetic data generated by numerically integrating (16) from initial conditions uniformly sampled from the region $[-2, 2]^3$. The model is implemented as a conditional DDPM with a U-Net architecture featuring 4 downsampling blocks, each with 64, 128, 256, and 512 channels respectively, trained for 1000 epochs with a cosine noise schedule. As we seek adherence to the true dynamics (16), we define the barrier function as:

$$h(x) = e - \frac{1}{L} \sum_{l=0}^{L-1} \left\| \frac{z^{l+1} - z^l}{\Delta_l} - \dot{z}^l \right\|^2 \tag{17}$$

where $\dot{z}^l$ is given by the true vector field (16), and $e$ is an error tolerance in the average physics adherence. The safe set $\mathcal{C} = \{x \mid h(x) \geq 0\}$ enforces the physics up to a tolerance $e = 0.001$.

Figure 2 presents key results of our CBF-guided sampling for Lorenz system trajectory generation. Figure 2(a) shows trajectories in the phase plane. The true ODE solution (dashed black) is closely tracked by our CBF-guided diffusion model (blue), while unconstrained sampling (red) produces trajectories that deviate significantly from the true physics. The unconstrained sample still produces the characteristic butterfly effect, but it is not the true trajectory that is followed by the system. This arises due to the fact that the diffusion model has learned to produce samples that exhibit the butterfly attractor by seeing ground truth data in training. This highlights a subtle but critical failure mode: a generative model can produce samples that are statistically indistinguishable from real data yet physically incorrect, and any downstream task that consumes such samples, whether a controller, a simulator, or a decision-making system, would operate on corrupted inputs without any indication of the violation.

Figure 2(b) reveals the evolution of the safety tube. At $t = 1$, the noise sample $x(T) \sim \mathcal{N}(0, I)$ has large cumulative physics error, requiring relaxation $\epsilon_0 \approx 16$—over four orders of magnitude larger than our target tolerance $e = 0.001$. The blue line shows the linear constriction $\epsilon(x(T), t) = \epsilon_0 \cdot (t/T)$ to zero at $t = 0$. The constriction barrier $\tilde{h}(x, t)$ (red) remains non-negative throughout, verifying reverse invariance (Theorem 4.2). Fluctuations during $t \approx 0.8$-$1.0$ reflect the high-noise regime where $g(t)$ is large. At $t = 0$, $\tilde{h}(x, 0) \approx 0$ with $\epsilon(x(T), 0) = 0$ implies $h(x(0)) \approx 0$, confirming that our minimum-norm control maximally exploits the constraint tolerance. Figure 2(c) shows control effort $\|u\|^2$ concentrated at sampling onset ($t \approx 0.8$-$1.0$), peaking around 130-140, then rapidly decaying to near-zero by $t \approx 0.6$. This front-loaded intervention exploits that control is cheaper when $g(t)$ is large. The minimal effort for $t < 0.6$ indicates that the pre-trained model has implicitly learned the Lorenz system structure from training data, requiring only minor corrections to enforce hard guarantees.

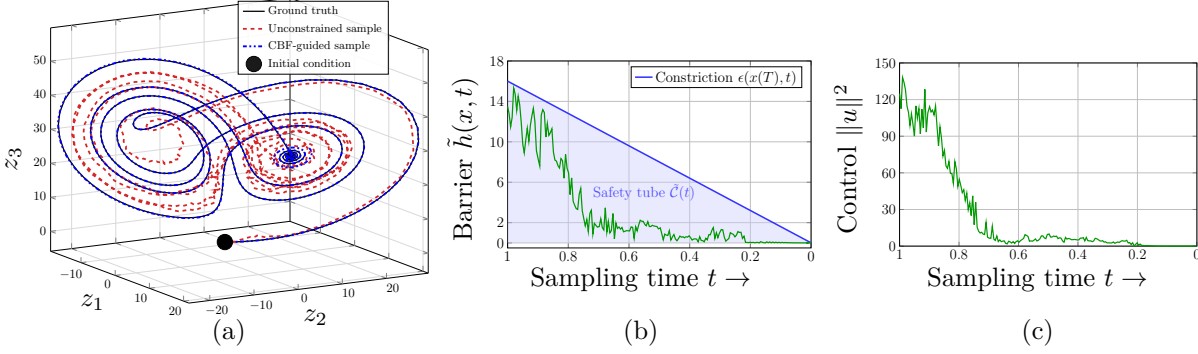

Figure 2: Lorenz system trajectory generation with CBF guidance. (a) Phase portrait showing the true ODE solution (dashed black). Our CBF-guided diffusion model (blue) closely tracks the true dynamics, while unconstrained sampling (red) produces physically inconsistent trajectories. (b) Evolution of the constricting barrier $\tilde{h}(x, t)$ (green) and relaxation $\epsilon(x(T), t)$ (blue) during sampling. The constriction goes from $\epsilon_0 \approx 16$ to 0, accommodating the large initial physics violation. The barrier remains non-negative throughout, verifying reverse invariance. (c) Control effort $\|u\|^2$ over sampling time, concentrated at the onset of sampling when the initial noise sample must be corrected, then rapidly diminishing as the trajectory becomes physics-consistent.

## 5.2 Constrained image generation

We demonstrate our framework on image synthesis tasks using the off-the-shelf DDPM-bedroom-256 [4] model from Hugging Face Diffusers, trained on the LSUN bedroom dataset (Yu et al., 2015). Images are represented as $x \in \mathbb{R}^{256 \times 256 \times 3}$ with RGB values normalized to $[-1, 1]$.

A key feature of our framework in this setting is that we define one barrier function per constrained pixel, rather than a single aggregate barrier over all pixels. For each pixel $\mathbf{p}$ in the constrained region, we define an independent barrier $h_{\mathbf{p}}(x)$ and enforce $h_{\mathbf{p}}(x) \geq 0$ separately. The resulting QP at each sampling step contains $|R|$ linear constraints, where $R$ is the set of constrained pixels. Crucially, since each per-pixel barrier $h_{\mathbf{p}}$ depends only on the three RGB values $x_{\mathbf{p}} \in \mathbb{R}^3$ at pixel $\mathbf{p}$. This sparsity implies that the multi-constraint QP over the full image space $\mathbb{R}^{256 \times 256 \times 3}$ decomposes into $|\mathcal{R}|$ independent three-dimensional QPs, one per pixel. Each sub-problem admits the closed-form solution from Remark 2, enabling efficient parallel computation. In our experiments, we solve the full QP using OSQP via CVXPY, which exploits this sparsity internally.

### 5.2.1 Location and content constraints

In this experiment, we enforce specific visual content at designated pixel locations, while retaining semantic meaning with the rest of the image. Given any reference image, we constrain a rectangular region $R \subseteq \mathbb{N}^2$ of dimension $h \times w$ in the generated image $x \in \mathbb{R}^{256 \times 256 \times 3}$ to match the reference.

To ensure the reference image appears in the prescribed location, we enforce pixel-level constraints inside the region $R$. Further, we modulate the constraint strength near boundary of $R$ so that the diffusion model can smoothly fill the edges of the image with semantic information. We define a spatially-varying mask function $v : \mathbb{N}^2 \to [0, 1]$ for each pixel $\mathbf{p} \in \mathbb{R}^{256 \times 256 \times 3}$:

$$v(\mathbf{p}) = \begin{cases} 1 & \text{if } \mathbf{p} \in \text{interior}(R) \\ \text{smooth decay to } 0 & \text{if } \mathbf{p} \in \text{boundary}(R) \\ 0 & \text{if } \mathbf{p} \notin R \end{cases}$$

where the boundary region is defined as pixels within 5% of the edge of $R$, and the decay is implemented via a linear ramp. Let $x_{\mathbf{p}}^* \in \mathbb{R}^3$ denote the RGB values of the reference image $\mathbf{P}$ at the corresponding position within $R$. For each pixel $\mathbf{p} \in R$, we define the barrier function:

$$h_{\mathbf{p}}(x) = e - v(\mathbf{p}) \cdot \|x_{\mathbf{p}} - x_{\mathbf{p}}^*\|^2, \tag{18}$$

where $e = 0.005$ is the error tolerance and $x_{\mathbf{p}} \in \mathbb{R}^3$ denotes the RGB pixel values of $x$ at location $\mathbf{p}$. The safe set is the intersection $\mathcal{C} = \bigcap_{\mathbf{p} \in R} \{x \mid h_{\mathbf{p}}(x) \geq 0\}$, which enforces pixel-level fidelity in the interior of $R$ where $v(\mathbf{p}) = 1$, while allowing deviations near boundaries where $v(\mathbf{p}) < 1$. The smooth decay $v(\mathbf{p}) \to 0$ allows the diffusion model to alter the edges of the reference image $\mathbf{P}$ as required for natural blending. Each pixel has its own constricting barrier $\tilde{h}_{\mathbf{p}}(x, t) = h_{\mathbf{p}}(x) + \epsilon_{\mathbf{p}}(x(T), t)$ with a linear relaxation $\epsilon_{\mathbf{p}}(x(T), t) = \epsilon_{0,\mathbf{p}} \cdot t/T$, where $\epsilon_{0,\mathbf{p}} = \max(0, -h_{\mathbf{p}}(x(T))) + c$, where $c = 0.01$ is a small positive margin.

Figure 3 demonstrates constrained image generation with a $50 \times 70$ pixel window region placed at position $(40, 150)$. Both generated images (b-c) preserve the reference window (a) exactly, validating Theorem 4.2. With 200 sampling steps (c), the model generates a coherent bedroom scene with semantically appropriate context—bed, lamps, and furniture properly scaled and lit relative to the window. The spatially-varying mask enables smooth blending at boundaries without visible artifacts. With only 50 sampling steps (b), visual quality degrades in unconstrained regions, yet the window remains perfectly preserved. This demonstrates that the CBF shield guarantees constraint satisfaction regardless of sampling duration, while unconstrained regions follow standard diffusion model behavior. This arises due to how the constriction $\epsilon$ is constructed with conditions in Definition 1.

This experiment demonstrates the modularity of our framework. We apply our method to an off-the-shelf pre-trained model without any architectural modifications or retraining, enforcing hard spatial constraints that would be impossible to guarantee with soft guidance methods. Additional results with different reference images and placements are provided in Appendix A.1.

---

[4] https://huggingface.co/google/ddpm-bedroom-256

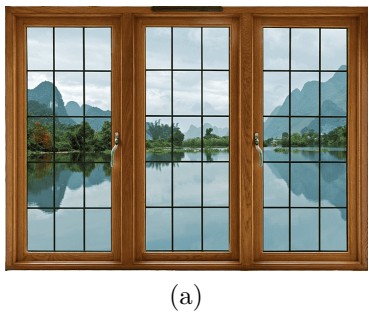 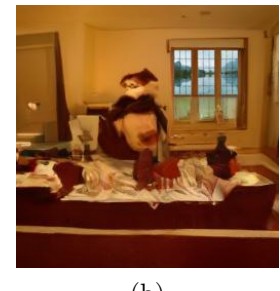 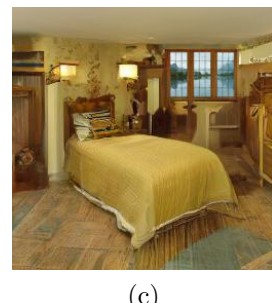

(a)                          (b)                          (c)

Figure 3: Spatially constrained image generation with DDPM-bedroom-256. (a) Reference window patch ($50 \times 70$ pixels). (b) Generated image with 50 sampling steps. (c) Generated image with 200 sampling steps. Both (b) and (c) preserve the reference window exactly at the specified location while generating bedroom context. Higher sampling steps improve generation quality in unconstrained regions, but constraint satisfaction is guaranteed in both cases.

### 5.2.2 Regional color intensity constraints

We demonstrate enforcement of regional color constraints by constraining the lower one-third of the image to maintain specified color intensities. Define the constrained region as:

$$R_{\text{lower}} = \{(i,j) \in \mathbb{N}^2 : 171 \le i \le 256, \ 1 \le j \le 256\},$$

which corresponds to the lower one-third of the image. The target color intensity is $x_{\mathbf{p}}^* \in [-1,1]^3$ for all $\mathbf{p} \in R_{\text{lower}}$. To allow natural blending near the boundary, we employ a spatially varying mask function $v : \mathbb{N}^2 \to [0,1]$ that modulates constraint strength:

$$v(\mathbf{p}) = \begin{cases} \frac{i-i_{\min}}{i_{\max}-i_{\min}} \cdot v_{\max} + \left(1 - \frac{i-i_{\min}}{i_{\max}-i_{\min}}\right) \cdot v_{\min}, & \text{if } \mathbf{p} = (i,j) \in R_{\text{lower}}, \\ 0, & \text{otherwise,} \end{cases} \tag{19}$$

where $i_{\min} = 171$, $i_{\max} = 256$ denote the row boundaries of $R_{\text{lower}}$, and $(v_{\min}, v_{\max}) \in [0,1]^2$ the constraint strength, respectively. For each pixel $\mathbf{p} \in R_{\text{lower}}$, the barrier function is:

$$h_{\mathbf{p}}(x) = e - v(\mathbf{p}) \cdot \|x_{\mathbf{p}} - x_{\mathbf{p}}^*\|^2, \tag{20}$$

where $e = 0.05$ is the error tolerance, and $x_{\mathbf{p}} \in \mathbb{R}^3$ denotes the RGB pixel values of $x$ at pixel $\mathbf{p}$. The safe set $C = \{x \in \mathbb{R}^{256 \times 256 \times 3} \mid h(x) \ge 0\}$ represents images with pixel-level color fidelity weighted by the mask function $v(\mathbf{p})$, where higher mask values enforce stricter adherence to the target intensity. We use the linear constriction scheme $\epsilon(x(T), t) = \epsilon_0 \cdot t/T + c$ where $\epsilon_0 = \max(0, -h(x(T)))$.

In this experiment, we use different mask configurations and color intensities and compare our constricting CBF guidance scheme with projection-based constraint enforcement in Zampini et al. (2025). This experiment illustrates how our coarse-to-fine constraint enforcement retains semantic meaning with the rest of the image whereas projection schemes can lose semantic meaning:

1. *Moderate constraint* in Figure 4(a): The target pixel color is black, $x_{\mathbf{p}}^* = (-0.9, -0.9, -0.9)$, and the spatial mask $v(\mathbf{p})$ varies from $v_{\max} = 0.5$ at the bottom to $v_{\min} = 0$ at the upper boundary of $R_{\text{lower}}$. This allows the diffusion model freedom to generate detailed, realistic textures while satisfying the per-pixel color constraints.

2. *Projection* in Figure 4(b): We implement the projection-based constraint enforcement scheme in Zampini et al. (2025) and compare the visual quality. The constraint is the same as in the earlier case where the bottom-third of the image needs to be black $x_{\mathbf{p}}^* = (-0.9, -0.9, -0.9)$, and the spatial mask $v(\mathbf{p})$ varies from $v_{\max} = 0.5$ at the bottom to $v_{\min} = 0$ at the upper boundary of $R_{\text{lower}}$. Due to projection at each step, we get a black-tape effect, which achieves constraint enforcement but semantic meaning is lost.

3. *Weak constraint* in Figure 4(c): The target pixel intensity is $x_{\mathbf{p}} = (-0.4, -0.3, -0.3)$, corresponding to a light brown color. The mask values range from $(v_{\min}, v_{\max}) = (0, 0.2)$, applying weak constraints. This allows the diffusion model to almost freely generate an image, relying on the guidance to only nudge individual pixels toward the target color.

All images successfully satisfy the color constraint in the lower region while maintaining semantic coherence with realistic bedroom layouts. However, the choice of mask strength directly affects the balance between strict constraint adherence and natural visual appearance, with lower mask values allowing greater model freedom to generate detailed, realistic textures within the constrained color range. Figure 4 illustrates how our CBF-guided sampling framework enables precise control over the strength and extent of constraint enforcement. The spatially-varying mask function $v(p)$ provides a smooth control between strict enforcement and model freedom, allowing practitioners to balance safety enforcement with perceptual quality based on application requirements. Importantly, all three configurations achieve 100% constraint satisfaction. No generated image violates $h(x) \geq 0$, demonstrating the formal safety guarantees of Theorem 4.2 even under weak constraint settings.

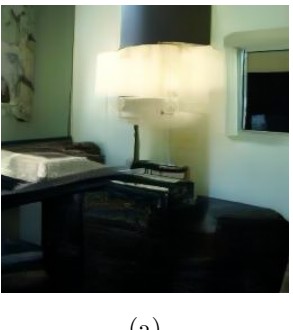 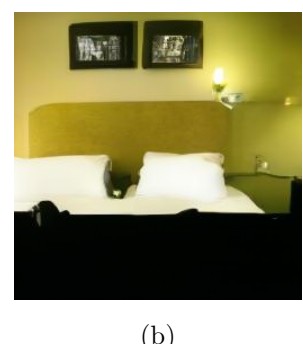 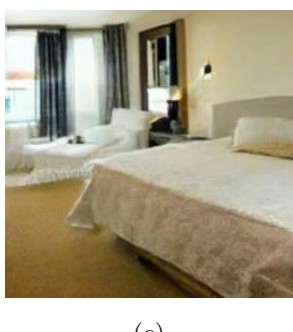

(a)  (b)  (c)

Figure 4: Regional color intensity constraints. Images are enforced with color values in the lower third. (a) *Moderate intensity:* The target pixel color is black, and the spatial mask $v(\mathbf{p})$ goes from $0.5 \to 0$, allowing the diffusion model to produce clean furniture. (b) *Projection:* Projection-based constraint enforcement (Zampini et al., 2025), constraints are the same as in (a), with the target pixel color black in the bottom one-third of the image. Projection causes intense constraint enforcement, which leads to loss of semantic information. (c) *Low intensity:* Target color is brown, and the spatial mask $v(\mathbf{p})$ goes from 0.2 at the bottom to 0 at the lower-third boundary, producing a semantically meaningful image of a room with a brown carpet.

### 5.2.3 Quantitative study and ablation: choice of constriction schedule

We now study how the choice of constriction schedule $\epsilon(x(T), t)$ affects sampling. As shown in Section 4.1, Definition 1 admits a broad family of schedules, and we proposed three concrete candidates: *linear*, *exponential*, and *polynomial*. The exponential schedule with $\lambda > 1$ front-loads constriction into the high-noise regime, the polynomial schedule with $p > 1$ back-loads it, and the linear schedule applies uniform pressure throughout. To validate this analytical prediction, we run the location and content constraint of Section 5.2.1 under four schedules with $K = 200$ sampling steps and identical safety margin $c = 0.01$. We compare:

- **Linear**: $\epsilon(x(T), t) = \epsilon_0 \cdot t/T$.

- **Exponential, $\lambda = 1.5$**: $\epsilon(x(T), t) = \epsilon_0 \cdot (e^{\lambda t/T} - 1)/(e^{\lambda} - 1)$, mild front-loading.

- **Exponential, $\lambda = 3$**: same form, aggressive front-loading.

- **Polynomial, $p = 3$**: $\epsilon(x(T), t) = \epsilon_0 \cdot (t/T)^p$, back-loading.

We measure two complementary aspects of each schedule. To characterize per-step control behavior, we run sampling with a single fixed initial noise $x(T)$ and log the maximum per-step linearization error

$\max_k |\tilde{h}(x_{k-1}, t_{k-1}) - \tilde{h}_{\mathrm{linear},k}|$, where $\tilde{h}_{\mathrm{linear},k}$ is the prediction implied by the linearized constraint at step $k$. To assess quality and diversity of the generated distribution, we generate 500 independent samples per schedule (varying random seed) and compare against 500 samples from the unguided model using Fréchet Inception Distance (FID), Kernel Inception Distance (KID), and Vendi score (Friedman & Dieng, 2023). As the constraint intentionally alters the distribution within the window patch, we compute FID and Vendi on the *masked* variant of each image, in which the constrained region is replaced with a uniform fill. This isolates whether guidance preserves the model's expressiveness in regions where the constraint is inactive. Aggregate results are reported in Table 1 and Figure 5.

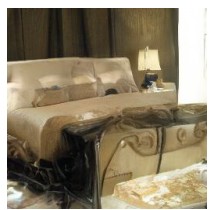 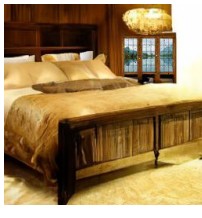 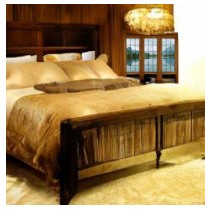 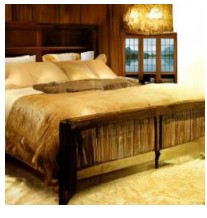 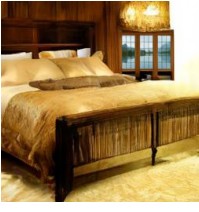

| (a) Unconstrained | (b) Linear | (c) Exp, $\lambda = 1.5$ | (d) Exp, $\lambda = 3$ | (e) Poly, $p = 3$ |

Figure 5: Final samples comparing sampling with an unconstrained model versus four constriction schedules on the window-location constraint of Section 5.2.1 under identical initial noise. All four schedules preserve the reference window patch. The schedules differ in where along the sampling horizon the control effort is concentrated. See Table 1 for quantitative comparison.

Table 1: Quantitative comparison of constriction schedules on the location and content constraint, $K = 200$ sampling steps. $\max_k \mathrm{lin.err}_k$ is the largest per-step deviation between the realized barrier and its linear prediction (lower is better, characterizes discrete-time error). $\mathrm{FID}_{\mathrm{mask}}$ and $\mathrm{KID}_{\mathrm{mask}}$ are computed between 500 schedule samples and 500 vanilla samples on the unconstrained region only (lower indicates closer to vanilla). $\mathrm{Vendi}_{\mathrm{mask}}$ is intra-set diversity on the unconstrained region (higher is more diverse, vanilla reference is 7.35). QP time is the average per-step computation time for the QP.

| Schedule | $\max_k \mathrm{lin.err}_k$ | $\mathrm{FID}_{\mathrm{mask}}$ | $\mathrm{KID}_{\mathrm{mask}}$ ($\times 10^{-2}$) | $\mathrm{Vendi}_{\mathrm{mask}}$ | QP time (ms) |
|---|---|---|---|---|---|
| Vanilla (no CBF) | – | – | – | 7.35 | – |
| Linear | 7.42 | 47.09 | $1.01 \pm 0.19$ | 6.62 | 33.45 |
| Exponential, $\lambda = 1.5$ | 10.04 | 47.22 | $1.03 \pm 0.19$ | 6.64 | 32.16 |
| Exponential, $\lambda = 3$ | **5.81** | 47.22 | $1.04 \pm 0.19$ | **6.66** | 33.62 |
| Polynomial, $p = 3$ | 22.69 | 47.20 | $1.04 \pm 0.19$ | 6.65 | 33.21 |

Three observations emerge from Table 1 and Figure 5.

*(i) The aggressive exponential schedule minimizes linearization error.* The maximum per-step linearization error for $\lambda = 3$ (5.81) is the smallest among all schedules. As the linearization (14) is a first-order Taylor expansion of $\tilde{h}$ around the current state, its accuracy depends on the per-step change in $\tilde{h}$, which is dominated by the constriction term $\partial \epsilon / \partial t \cdot \Delta t$ when the QP is active. By concentrating $\partial \epsilon / \partial t$ in the high-noise regime where the underlying stochastic dynamics already dominate the per-step state change, the front-loaded schedule keeps the constriction-induced linearization residual small throughout sampling. This is consistent with the prediction of Theorem 4.3: the factor $1/g(t)^2$ in (13) makes intervention distributionally cheap precisely where the noise level $g(t)$ is large, which is also where the Taylor expansion is least strained. The mild exponential ($\lambda = 1.5$) does not concentrate constriction aggressively enough to gain this advantage and lands between linear and aggressive exponential on every metric.

*(ii) The back-loaded polynomial schedule incurs higher linearization error.* The polynomial schedule with $p = 3$ exhibits a maximum linearization error of 22.69, nearly four times that of $\lambda = 3$. This is consistent with the inverse perspective on Theorem 4.3: by concentrating $\partial \epsilon / \partial t$ in the low-noise regime where the stochastic dynamics are quiescent, the back-loaded schedule forces the QP to make abrupt corrections that

strain the first-order approximation. Although this does not prevent successful sampling, it suggests that back-loaded schedules carry a numerical fragility that front-loaded schedules avoid.

*(iii) Generation quality on the unconstrained region is preserved and is essentially independent of schedule.* $\text{FID}_{\text{mask}}$ varies only between 47.09 and 47.22 across all four schedules, differences well within the KID standard error of $\pm 0.19 \times 10^{-2}$. $\text{Vendi}_{\text{mask}}$ similarly varies only between 6.62 and 6.66, against a vanilla reference of 7.35, indicating that all CBF schedules preserve approximately 90% of the unguided model's diversity in the unconstrained region. The remaining 10% reduction is attributable to the fact that all 500 samples share the same forced window content, biasing the model toward a narrower distribution of compatible scenes rather than reflecting damage to its generative capability. This is a useful negative result for practitioners: the choice of schedule can be made on control-theoretic grounds (linearization stability) without trading off generation quality.

Across all CBF schedules, the per-sample QP overhead is approximately 33 ms, roughly 13% relative to vanilla sampling. The remaining inference time is dominated by U-Net forward passes, which are unaffected by the choice of constriction schedule. The front-loaded exponential schedule with $\lambda \geq 3$ is a robust default: it minimizes maximum linearization error while preserving unconstrained-region quality on par with the alternatives. Linear and back-loaded schedules remain valid and satisfy the safety guarantee of Theorem 4.2, but the discrete-time implementation will be most robust when constriction is concentrated where the model itself has not yet committed to fine structure.

### 5.3 Smooth robot policy generation

We demonstrate our framework on robotic manipulation using the pre-trained Diffusion Policy model (Chi et al., 2025) for the Push-T task. The Push-T task requires a planar robotic arm to push a T-shaped block to a target pose, as depicted in Figure 6(a). The diffusion policy generates a sequence of waypoints $x(0) = \{a_0, a_1, \ldots, a_S\}$, where $S = 15$ is the action horizon and each waypoint $a_s \in \mathbb{R}^2$ represents a position that is tracked by a low-level controller. We seek safe samples of action chunks $x(0)$ by applying Algorithm 1 during the diffusion sampling process, without any retraining or modification to the model architecture. On physical robotic hardware, abrupt changes in waypoints pose concrete safety risks. Large accelerations violate actuator rate limits, inducing torque spikes that can damage motors and gearboxes. In contact-rich tasks such as Push-T, jerky motions destabilize the grasp or push contact, leading to task failure or uncontrolled object motion. Smoothness of the commanded action sequence is therefore not merely a quality metric but an operational safety requirement for real-world deployment. We encode this requirement as a hard constraint by bounding the average curvature variation across the action horizon. Specifically, we define the barrier function as:

$$h(x) = e_{\text{smooth}} - \frac{1}{S} \sum_{s=1}^{S-1} \|a_{s+1} - 2a_s + a_{s-1}\|^2, \tag{21}$$

where $e_{\text{smooth}} = 1.5$ is the tolerance that bounds the maximum allowed average curvature. The safe set $\mathcal{C} = \{x \mid h(x) \geq 0\}$ encodes the set of all action sequences whose average curvature remains below the prescribed threshold $e_{\text{smooth}}$. We apply our Algorithm 1 with the same linear constriction scheme as in the previous experiments.

Table 2 presents quantitative results averaged over 100 episodes. We compare our CBF-guided sampling against the original Diffusion Policy with DDPM sampling (DP) and DDIM sampling (DP-DDIM), using the pre-trained checkpoints from the authors' repository[5]. Our CBF-guided sampling achieves a mean reward of 0.92, matching the original Diffusion Policy, while guaranteeing zero smoothness violations across all episodes. In contrast, unconstrained DP and DP-DDIM exceed the smoothness threshold $\epsilon_{\text{smooth}}$ an average of 14 and 19 times per episode, respectively. This demonstrates that the learned policy does not inherently satisfy smoothness requirements despite being trained on smooth demonstrations. The diffusion sampling process itself introduces high-frequency artifacts that violate the constraint. Figure 6(b) visualizes the end-effector trajectory for a representative episode. The unconstrained DP trajectory (dotted red) exhibits

---

[5]https://github.com/real-stanford/diffusion_policy

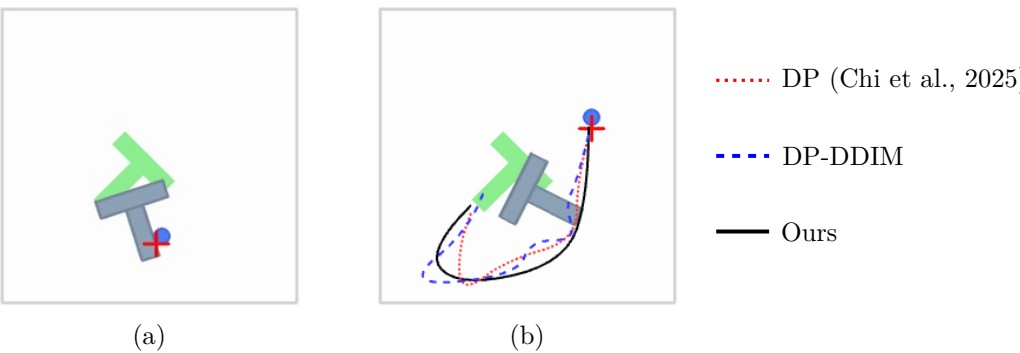

(a)          (b)

Figure 6: Smooth action generation for Push-T robotic manipulation. (a) The Push-T environment: a planar robot arm must push the T-shaped block (gray) to the target pose (green). (b) End-effector trajectories for a representative episode. Unconstrained Diffusion Policy (DP, dotted red) exhibits sharp directional changes during pushing. DP-DDIM (dashed blue) produces more erratic motion due to fewer sampling steps. Our CBF-guided sampling (solid black) generates a smooth trajectory that satisfies the smoothness constraint at every step while achieving the same task reward.

Table 2: Comparison on Push-T with smoothness constraints over 100 episodes

| Metric | DP | DP-DDIM | CBF-guided |
|---|---|---|---|
| Mean reward | 0.92 | 0.90 | 0.92 |
| Smoothness violation | 14 | 19 | **0** |
| Sampling steps $K$ | 100 | 10 | 100 |
| Mean inference time (ms) | 47.05 | 4.57 | 62.91 |

sharp directional changes, particularly during the pushing phase. DP-DDIM (dashed blue), which uses fewer sampling steps, produces even more erratic motion. Our CBF-guided trajectory (black) follows a visibly smoother path while achieving the same task objective. The smoothness is enforced throughout the diffusion sampling process rather than applied as post-hoc filtering, ensuring that every generated action chunk satisfies the constraint by construction. The additional computational cost of solving the QP at each sampling step is modest. Inference time increases from 47 ms to 63 ms per sample, a 34% overhead that remains well within real-time requirements for the 10 Hz control loop of the Push-T environment. DP-DDIM achieves significantly faster inference (4.6 ms) by using only 10 sampling steps, but at the cost of higher smoothness violations and slightly lower task reward.

This experiment validates our framework in a robotic planning setting where the generative model operates over structured action sequences rather than images or physical trajectories. The constraint is enforced purely in action space, requiring no forward dynamics model or state prediction. Extending to state-space safety constraints, such as collision avoidance, would require access to a dynamics model (analytical or learned) to propagate the effect of actions to states. We discuss this as a natural extension in Section 6.

## 6 Limitations and future work

Our framework assumes access to a continuously differentiable barrier function $h(x)$ that precisely encodes the safety specification. When the notion of safety is well-defined and analytically expressible, such as physics residuals (Section 5.1), pixel-level constraints (Section 5.2), or action smoothness bounds (Section 5.3), our method provides deterministic guarantees. However, in domains where safety is ambiguous or difficult to formalize, such as filtering semantically inappropriate content in images, constructing a suitable barrier is nontrivial. In our preliminary experiments, we attempted to use CLIP-based confidence scores and neural network classifiers as barrier functions for semantic constraints. These attempts were unsuccessful on several

occasions. The classifiers occasionally assigned high confidence to unsafe samples, causing the CBF condition to be trivially satisfied and the safety filter to remain inactive. This failure mode is well-documented in the adversarial robustness literature and stems from the fact that learned classifiers extract features that do not reliably align with the safety boundary. When the barrier function itself is unreliable, obtaining formal guarantees on safety with respect to the intended safety semantics is unachievable.

Several extensions follow naturally from our framework. First, extending the approach to latent diffusion models would enable scalability to higher-resolution generation. We investigated extending our framework to latent diffusion models (Rombach et al., 2022) where the diffusion process operates in a compressed latent space $z = E(x)$ and the constraint is defined in image space. While the QP structure is preserved, the latent gradient is computed via a single backward pass through the decoder. Our preliminary experiments with Stable Diffusion v1.5 revealed that exact constraint satisfaction does not transfer from latent to image space (see Appendix B.2). The VAE decoder is not a diffeomorphism: it is locally non-invertible and introduces reconstruction artifacts that distort the constraint boundary. As a result, the CBF shield successfully biases the latent trajectory toward the constrained region but cannot guarantee pixel-level fidelity in the decoded image. Achieving exact constraints in latent diffusion models likely requires either (i) a constraint-aware fine-tuning of the decoder to improve local invertibility near the constraint boundary, or (ii) a hybrid approach that applies coarse guidance in latent space and a final correction step in pixel space after decoding. Second, enforcing state-space safety constraints in robotic policies, such as collision avoidance, requires a dynamics model to map actions to states. Combining our framework with learned dynamics models or neural ODEs is a promising direction. Third, our current implementation solves one QP per sampling step using a general-purpose solver. Using an MPC framework across many steps could significantly yield tighter bounds on the KL divergence over the entire sampling horizon. In particular, a receding-horizon formulation that anticipates the constriction schedule over multiple future steps could yield control policies that are globally optimal, rather than greedy at each step, potentially tightening the bound over the learned sampling process.

## 7 Conclusion

We introduced a framework for enforcing hard constraints on flow-based generative models by framing guided sampling as a problem of control synthesis. Our approach leverages constricting Control Barrier Functions (CBFs) to define a safety tube that is relaxed at the initial noise distribution and progressively constricts to the target safe set. The resulting minimum-norm QP synthesizes a feedback control input that provably maintains samples within the safety tube (Theorem 4.2), while minimizing the instantaneous contribution to the distributional shift from the learned model, as quantified by the KL divergence (Theorem 4.3). We validated the modularity and effectiveness of this framework across physically-consistent trajectory generation, constrained image synthesis, and smooth action generation for robotic manipulation, using off-the-shelf pre-trained models. Our framework maintains semantic fidelity of the model while ensuring constraint adherence. As generative models are increasingly deployed in safety-critical systems, this framework provides a principled safety layer that complements the expressiveness of generative sampling models.

## Broader Impact Statement

Our framework is motivated by safety: it provides a deterministic mechanism for enforcing hard constraints on pre-trained generative models, which we expect to be most useful in safety-critical settings such as robotics and scientific simulation. The same mechanism that enforces benign constraints could in principle be used to force generative models to satisfy harmful or deceptive specifications, and this concern grows as the approach is extended toward latent-space and high-resolution image and video models, where constraints could be used to embed targeted or undesirable content. We believe the net effect is positive, which is that hard, auditable constraints are easier to inspect and regulate rather than opaque soft guidance.

We also stress an important limitation of the guarantee itself. Our safety certificate is only as meaningful as the barrier function that encodes it: the theorem guarantees membership in the set $\{x : h(x) \geq 0\}$, not that this set captures the intended notion of safety. When the barrier is misspecified, or when it is a learned proxy (e.g., a classifier) that does not reliably align with the true safety boundary, the formal guarantee

can hold while the intended safety property fails, a failure mode we observed directly in our preliminary experiments (Section 6). Practitioners should therefore treat our framework as a way to enforce a *specified* constraint exactly, not as a substitute for specifying what safety means.

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

# A Appendix: Additional experiments

## A.1 Spatial content constraints with alternate reference images

Figure 7 demonstrates spatial and content-constrained image generation using red decorative pillows as the reference patch. Both constrained generations (b-c) preserve the pillows exactly at the specified location.

Notably, the generated scenes exhibit predominantly light tones. This occurs because the reference patch (a) includes white pixels near its boundaries, which the diffusion model interprets as local context. The spatially-varying mask $v(\mathbf{p})$ constrains boundary pixels to remain close to reference values, causing the model to generate ambient colors that transition naturally from the white boundary. This demonstrates that our CBF shield enforces constraints precisely as specified, including boundary characteristics. In practice, reference patches with neutral boundaries allow greater freedom in surrounding generation.

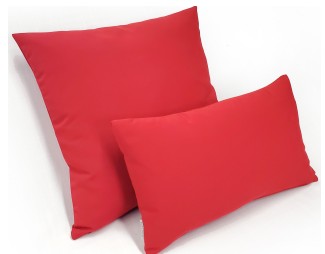 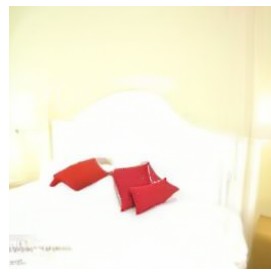 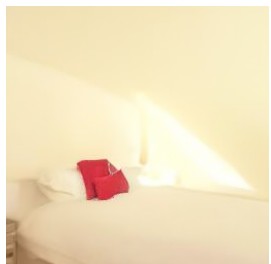

Reference 50 sampling steps 200 sampling steps

Figure 7: Spatial and content constrained generation with red pillow reference. (Left) Reference patch ($50 \times 50$ pixels) containing two red pillows with white background. (Middle) Generated image with 50 sampling steps. (Right) Generated image with 200 sampling steps. Both preserve the pillows exactly while generating bedroom context. The white boundary pixels in the reference influence the surrounding color palette, resulting in light-toned ambient environments.

## A.2 Non-convex constraints: disjunctive window selection

A central claim of our framework (Contribution 1) is that the safety guarantee of Theorem 4.2 makes no assumption on the convexity of the safe set. To demonstrate this directly in image space, we construct a constraint whose safe set is a *union* of two basins, an explicitly non-convex set, and show that the framework selects and renders one of them without any manual intervention.

Given two reference window images $A, B \in \mathbb{R}^{3 \times N}$ over the constrained patch of $N$ pixels, we require the patch to match *one of* the two references to within a tolerance $e$. Writing $A_p, B_p \in \mathbb{R}^3$ for the reference colors at pixel $p$ and $v(p) \in [0, 1]$ for the boundary mask of Section 5.2.1, the safe set is

$$\mathcal{C} = \left\{ x : \tfrac{1}{N} \sum_p v(p) \|x_p - A_p\|^2 \le e \right\} \cup \left\{ x : \tfrac{1}{N} \sum_p v(p) \|x_p - B_p\|^2 \le e \right\}. \tag{22}$$

This is the union of two disjoint basins (one per reference window) and is therefore non-convex: a patch interpolating between the two windows lies in neither basin, yet both endpoints are feasible.

The barrier for a union of safe sets is the pointwise maximum of the individual barriers, $h(x) = \max(h_A(x), h_B(x))$, where $h_A, h_B$ are the per-reference barriers. The hard maximum is non-differentiable on the set where $h_A = h_B$, and its gradient switches abruptly between the two references there, which would cause the control direction to chatter between basins across sampling steps. We therefore smooth the maximum with a LogSumExp surrogate,

$$h_\beta(x) = \tfrac{1}{\beta} \log\big(e^{\beta h_A(x)} + e^{\beta h_B(x)}\big), \tag{23}$$

whose gradient is a softmax-weighted blend of the two basin gradients, $\nabla h_\beta = \sigma_A \nabla h_A + \sigma_B \nabla h_B$ with $\sigma_A = e^{\beta h_A}/(e^{\beta h_A} + e^{\beta h_B})$. The surrogate is a $C^\infty$ approximation of the max function.

The control is synthesized per pixel against the selected target $T_p = \sigma_A A_p + \sigma_B B_p$, exactly as in Section 5.2.1. Because the two references occupy well-separated basins, $\sigma$ commits to one window ($\sigma \to 1$) within the first few sampling steps and remains committed, so $T_p$ is effectively the chosen reference. This design retains the per-pixel enforcement strength that renders a sharp window while keeping the non-convex two-basin choice at the aggregate level. We emphasize that the per-step QP remains convex: the maximum is resolved by evaluating $(\sigma_A, \sigma_B)$ at the current state, and the QP only ever sees a linear constraint.

Figure 8 shows generations under the disjunctive constraint at a fixed patch location, across several seeds, using the off-the-shelf DDPM-bedroom-256 model. The framework selects a window basin based solely on the initial noise and renders the selected window to within tolerance in every case. Across seeds, both windows are selected, confirming that both basins of the non-convex set are reachable and that the selection is genuinely determined by the sampling trajectory rather than fixed in advance.

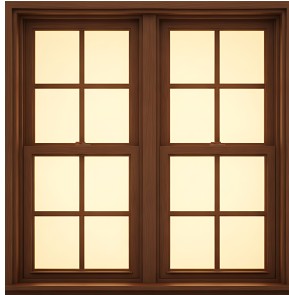 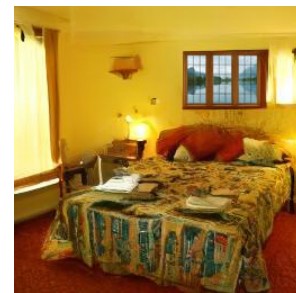 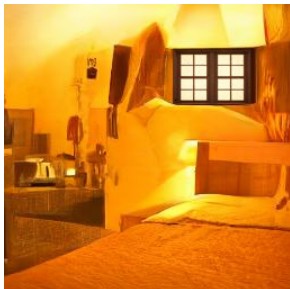

Figure 8: Disjunctive two-window constraint, a non-convex safe set {match window $A$} $\cup$ {match window $B$}. Window choice A is given in Figure 3(a), and window choice B is the *right* image. The constraint is applied at a fixed patch location (as in Section 5.2.1), only the random seed varies. The framework selects a reference window based on the initial noise, with no manual choice, and renders the selected window to within tolerance. Different seeds select different windows, confirming both basins of the non-convex set are individually reachable. Both reference windows are in-distribution for the pre-trained bedroom model, so the selected window integrates naturally into the surrounding scene.

This experiment isolates the non-convexity that Theorem 4.2 already admits in principle: the safe set is a union of basins, and the guidance mechanism navigates to one of them while preserving the safety certificate throughout sampling. We note that the experiment succeeds because both references are compatible with the generative prior (both are realistic windows that the bedroom model readily produces). When a reference is far from the data manifold, the constraint still admits a formal guarantee with respect to the specified barrier, but enforcement requires larger control effort and may degrade perceptual quality—consistent with the cooperation principle (Contribution 2): our framework is most effective when the constraint can be satisfied in a manner compatible with the model's learned structure.

## A.3 Ablation: cutoff steering control

A central claim of our framework is that the constricting tube concentrates constraint enforcement in the high-noise regime, where interventions are distributionally cheap, and that the control input vanishes ($u^* \to 0$) as the trajectory enters the low-noise regime near $t = 0$. Disabling the controller during the low-noise phase

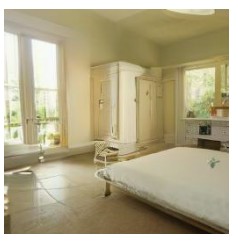 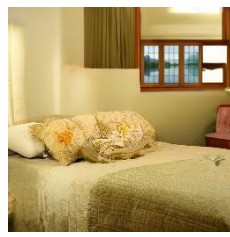 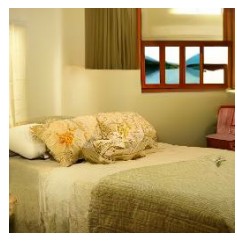 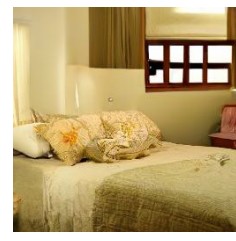 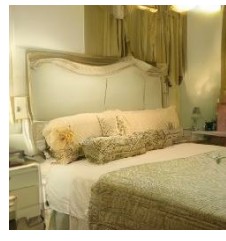

(a) unguided          (b) $t_{\text{off}} = 0.05$          (c) $t_{\text{off}} = 0.3$          (d) $t_{\text{off}} = 0.6$          (e) $t_{\text{off}} = 0.9$

Figure 9: Effect of the control cutoff (single fixed seed, linear constriction, spatial localization constraint of Section 5.2.1). Control is applied while $t > t_{\text{off}}$ and disabled for $t \leq t_{\text{off}}$. Sampling runs in reverse time from $t = T$ to $t = 0$, so a small $t_{\text{off}}$ keeps control active for almost the entire trajectory. **(a)** Unguided sampling: an unrelated scene with no patch. **(b)** $t_{\text{off}} = 0.05$: full fidelity, including the fine lines inside glass window, indistinguishable from always-on control. **(c)** $t_{\text{off}} = 0.3$: coarse content and placement preserved, but the finer glass details are lost. **(d)** $t_{\text{off}} = 0.6$: the window's coarse characteristics persist but the constraint is substantially violated. **(e)** $t_{\text{off}} = 0.9$: the constraint vanishes and the model reclaims the region. Control applied later in sampling enforces progressively finer scales of the constraint.

should have negligible effect on the final sample, whereas disabling it during the high-noise phase should cause the constraint to fail. We test this directly by truncating the control input at a release time $t_{\text{off}}$: the QP (15) is solved and applied while $t > t_{\text{off}}$, and the sampling proceeds unguided (vanilla) for $t \leq t_{\text{off}}$. Recall that sampling runs in reverse time from $t = T$ to $t = 0$. A small $t_{\text{off}}$ therefore means the controller remains active for almost the entire trajectory and is released only in the final steps, while a large $t_{\text{off}}$ means the controller is switched off early and the model samples freely through the entire refinement phase. We use the spatial localization constraint of Section 5.2.1 with a fixed window reference, a fixed location, a single fixed seed, and the linear constriction scheme, so that every condition shares the same noise realization $\{\xi(t)\}$ and the same initial relaxation $\epsilon_0$. The only variable is $t_{\text{off}}$. Figure 9 reports the resulting samples for $t_{\text{off}} \in \{0.05, 0.3, 0.6, 0.9\}$ alongside the fully unguided baseline.

The results reveal a coarse-to-fine degradation rather than a simple loss of the constraint. For $t_{\text{off}} = 0.05$, where the controller is released only in the final few low-noise steps, the constrained window patch is reproduced with full fidelity: the frame, the reflected landscape, and even the fine mullion lines across the glass are preserved, and the result is visually indistinguishable from the always-on case. Releasing this late costs essentially nothing because the tube has already tightened around the safe set and $u^* \approx 0$. At $t_{\text{off}} = 0.3$, the coarse content of the patch survives, but the fine mullion lines within the glass are lost, the individual panes merging into smoother fields. This is precisely the detail that the model resolves during the low-noise phase that we have now left uncontrolled, and its selective disappearance is direct evidence of the coarse-to-fine structure the tube is designed to exploit. At $t_{\text{off}} = 0.6$ the region still reads as a window, the dark frame and its placement persist, but the constraint is substantially violated: the landscape content is gone and the patch blends into the surrounding wall. By $t_{\text{off}} = 0.9$, where the controller acts only during the first few high-noise steps, the constraint vanishes entirely and the model reclaims the region with an unrelated headboard. The unguided baseline, with no control at any step, produces an unrelated scene with no patch. This ablation isolates *where along the sampling horizon* the steering secures each scale of the constraint. Fine, high-frequency detail is enforced only by control applied late in sampling. Coarse structure and placement are set earlier. Releasing during the high-noise phase loses the constraint details, but the structure of the window is retained. The progression is consistent with the invariance argument of Theorem 4.2, which requires the CBF condition to hold at every step down to $t = 0$ for exact satisfaction. Conversely, the indistinguishability of the $t_{\text{off}} = 0.05$ and always-on samples provides direct empirical support for the low-noise vanishing of $u^*$ predicted by the KL-divergence analysis of Theorem 4.3 and the $1/g(t)^2$ cost structure in (13): in the regime where intervention is most expensive in distributional terms, the controller has nothing left to do.

## A.4 Constrained image generation with latent diffusion models

We investigate extending our CBF framework to latent diffusion models (Rombach et al., 2022), where the diffusion process operates in a compressed latent space rather than directly in pixel space. This extension is motivated by the scalability requirements of modern text-to-image models such as Stable Diffusion, which perform sampling in a low-dimensional latent space $\mathbb{R}^{4\times64\times64}$ that is decoded to images in $\mathbb{R}^{3\times512\times512}$ via a variational autoencoder (VAE).

Let $E : \mathbb{R}^n \to \mathbb{R}^d$ and $D : \mathbb{R}^d \to \mathbb{R}^n$ denote the VAE encoder and decoder, respectively, where $d \ll n$. Given a constraint defined in image space via a barrier function $h : \mathbb{R}^n \to \mathbb{R}$, we define the *latent barrier function* as the composition

$$\bar{h}(z) := h(D(z)), \tag{24}$$

which lifts the image-space constraint into the latent space via the decoder. The constricting barrier becomes $\tilde{h}(z,t) = \bar{h}(z) + \epsilon(z(T), t)$, and the CBF condition for the latent dynamics $\mathrm{d}z = [f_\theta(z, t) + u]\mathrm{d}t + g(t)\mathrm{d}w$ requires the gradient

$$\nabla_z \bar{h}(z) = \left(\frac{\partial D}{\partial z}\right)^\top \nabla_x h(x)\bigg|_{x=D(z)}, \tag{25}$$

which is computed via a single backward pass through the decoder using automatic differentiation. The QP structure of Algorithm 1 is preserved, with the state space replaced by $\mathbb{R}^d$. We apply the spatial localization constraint from Section 5.2.1 to Stable Diffusion v1.5 (`sd-legacy/stable-diffusion-v1-5`) from Hugging Face Diffusers. The prompt to the model is "high quality image of a rustic bedroom". We constrain a rectangular region in the generated image to match a reference window patch, using the same per-pixel barrier function (18) evaluated on the decoded image $\hat{x} = D(z)$. The latent gradient (25) is backpropagated through the VAE decoder at each sampling step.

Figure 10 presents the results. The CBF-guided latent diffusion model responds to the spatial constraint: a visually distinct region appears at the specified location, indicating that the latent-space control input successfully biases the sampling trajectory toward the constrained region. However, the reference window patch is not preserved with pixel-level fidelity. The constraint region appears blurred, color-shifted, and blended into the surrounding scene, in contrast to the exact preservation achieved in pixel-space experiments (Figure 3 and 7). This degradation stems from a fundamental limitation that the VAE decoder $D$ is not a diffeomorphism. The mapping from latent to image space is locally non-invertible and many-to-one, meaning that multiple latent codes can decode to similar but non-identical images. Consequently, enforcing $\bar{h}(z) = h(D(z)) \geq 0$ in latent space does not guarantee that the decoded image $\hat{x} = D(z)$ satisfies $h(\hat{x}) \geq 0$ with the same precision. The decoder introduces reconstruction artifacts that distort the constraint boundary, causing the gradient $\nabla_z \bar{h}$ to point in directions that are approximately but not exactly aligned with the true image-space constraint.

Formally, the safety guarantee of Theorem 4.2 holds with respect to the composed barrier $\bar{h}(z) = h(D(z))$: if $\tilde{h}(z(0), 0) \geq 0$, then $h(D(z(0))) \geq 0$. The issue is that the safe set $\bar{\mathcal{C}} = \{z \mid h(D(z)) \geq 0\}$ in latent space does not correspond exactly to the intended safe set in image space when the decoder has non-trivial reconstruction error. This is not a failure of the CBF framework itself, but rather a limitation of the representation space in which the diffusion process operates.

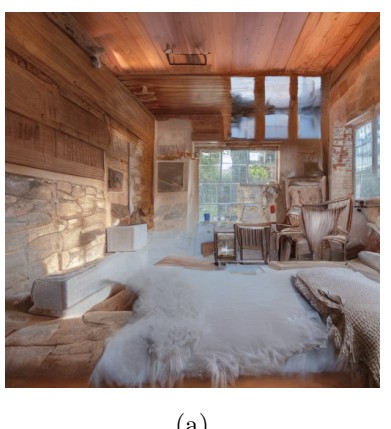 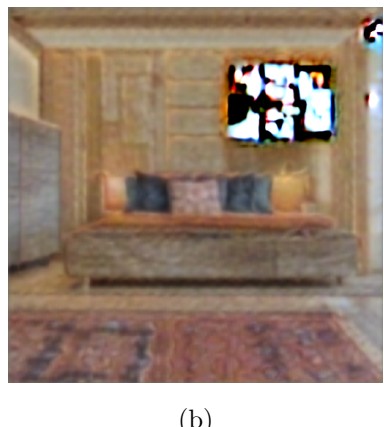

(a) (b)

Figure 10: Constrained image generation with latent diffusion models with prompt "high-quality image of a rustic bedroom". The constraint is to produce the reference window in Figure 3(a) in the top right corner, same as in Section 5.2.1. We see that the model responds to the constraint by producing a visually distinct region at the target location, but the reference patch is not preserved with pixel-level fidelity due to the VAE decoder's non-invertibility.

