# OpenReview forum: "Provably Safe Generative Sampling with Constricting Barrier Functions"
_TMLR — Decision pending for TMLR_

### Review · Reviewer_zgaV · 2026-04-13

**Summary Of Contributions:**

The paper proposes a sampling-time safety filter for flow-based generative models. Specifically, the paper defines a safety tube that is relaxed at the initial noise distribution and progressively tightens to the target safe set by the end of sampling. The proposed method has safety guarantees, minimizes the per-step contribution to the KL divergence between the safe and original distributions, and can be applied to any pre-trained flow-based generative model at sampling time without retraining or architectural modifications. The approach is validated on simulation of nonlinear physics, constrained image generation, safety-critical planning in robotics.

**Audience:**

Yes

**Audience Explanation:**

Enforcing constraints in generative models is an important problem across multiple domains. With experimental validation in physics simulation, image generation, and robotics, the paper will be of interest to researchers in both generative modeling and many downstream application domains.

**Claims And Evidence:**

No

**Claims Explanation:**

- The realized noise in the paper is defined as $\xi = dw/dt$, and is treated as an ordinary observed signal. However, a Wiener process $w$ has no pointwise derivative. This means that for Theorem 4.1, when the proof rewrites the stochastic system as if it were an ODE with known noise input, that step is not valid in the usual theory of stochastic processes.

- In the last part of the proof for Theorem 4.2, how does one go from the KL divergence between $Q$ and $P$ (Equation 23) to the KL divergence between $p_{safe}$ and $p_0$ (Equation 24)? The paragraph between these equations describes the derivation of the right hand side of the equation, which looks sound. However, the change in the left hand side is not justified.

- The theory provided covers only the continuous time framework, and it is not clear how much discrete time implementation will break the safety guarantee. Additionally, it is not clear how the conclusion that “the first-order Taylor expansion introduces an $O(\delta t^2)$ residual that is not formally bounded in our analysis” is reached.

- Experimental evaluations appear sound and exhaustive.

**Requested Changes:**

- Theorem 4.1 needs to either be restricted to a deterministic ODE setting, or a modified proof that handles the realized noise correctly.

- The last part of the proof for Theorem 4.2 needs more clarification. In particular, how does one go from the KL divergence between $Q$ and $P$ (Equation 23) to the KL divergence between $p_{safe}$ and $p_0$ (Equation 24)?

- As written, this paper does not prove exact discrete-time safety guarantees. This is relevant because practical algorithms, such as Algorithm 1, need to be discrete-time. If developing such a theory is challenging, the paper can at least substantially soften the title, abstract, and theorem language.

- (Not critical) It would be helpful to discuss whether the assumption “closed and bounded set C” usually holds.

- (Not critical) Would it be possible to expand on “In deterministic regimes g(t) = 0, such as flow matching, our framework naturally transitions into a minimal L2 drift perturbation” (page 7)? This seems to be an interesting observation, but was not fully formalized or proved in the paper.

---

> ### Author Response · Authors · 2026-05-29
> **Rigorous discrete-time guarantees and rescoped claims**
>
> - We thank the reviewer for flagging the definition of $\xi$ as a formal
>   derivative of a Wiener process, which has no pointwise derivative. Making the
>   continuous-time argument rigorous would require machinery such as the
>   Skorokhod problem for confined stochastic processes. Instead, we replace
>   Theorem 4.1 with a deterministic Proposition, use the analogous stochastic
>   system only to motivate the discretization, and make rigorous claims solely
>   in discrete time. Changes:
>     - Continuous-time deterministic invariance is now Proposition 4.1; for the
>       stochastic case we reference the Skorokhod problem as the route to full
>       continuous-time formalization, left to future work.
>     - The rigorous reverse-invariance result is now Theorem 4.2, stated and
>       proved at the Euler--Maruyama scheme the algorithm implements, by pathwise
>       induction on the discrete CBF condition over every noise realization. The
>       guarantee is exact in the discrete-time case for any noise realization.
>     - Section 3.1 now introduces the Euler--Maruyama discretization and index
>       convention explicitly.
>     - Section 3.2 now states the discrete CBF condition
>       $h(x_{k-1}) \geq (1 - \alpha \Delta t) h(x_k)$ alongside the continuous one.
>     - The footnote on $\xi$ is rewritten: $\xi$ is the realized noise observed
>       during simulation, with no reference to a derivative of $w$.
>     - The abstract, Contribution 1, and Conclusion now reflect the scoping:
>       rigorous discrete-time guarantees plus continuous-time motivation.
>   Algorithm 1 and all experiments are unaffected, as the algorithm operates on
>   observed discrete increments, exactly the object Theorem 4.2 certifies.
>
> - We have restructured the distribution-shift result entirely in discrete time,
>   and the step the reviewer identified is now an inequality (Theorem 4.3),
>   stated as a terminal-marginal bound. The proof has two stages. First, we bound
>   the path-measure divergence $D_{\mathrm{KL}}(Q_K \| P_K)$ between the joint
>   laws of the guided and unguided paths: conditioning on $(x_k, \xi_k)$ renders
>   $u_k$ deterministic, so the transitions are equal-covariance Gaussians
>   differing only by the mean shift $u_k\Delta t$, with per-step KL
>   $\tfrac{1}{2}\|u_k\|^2/g(t_k)^2 \cdot \Delta t$; the KL chain rule sums these.
>   Second (the flagged step), $p(0)$ and $p_{\mathrm{safe}}(0)$ are pushforwards
>   of $P_K, Q_K$ under $(x_K,\ldots,x_0) \mapsto x_0$, so the data-processing
>   inequality gives
>   $D_{\mathrm{KL}}(p_{\mathrm{safe}}(0)\|p(0)) \leq D_{\mathrm{KL}}(Q_K\|P_K)$.
>   The two are thus distinct quantities related by an inequality, resolving the
>   conflation in the original submission.
>
> - We have softened the continuous-time safety language while establishing
>   rigorous discrete-time guarantees: the invariance result is now discrete
>   (Theorem 4.2), with the abstract, Contribution 1, theorem statement, and
>   Conclusion updated accordingly. We also separate the exact discrete CBF
>   condition from the linearized QP (Section 4.3). The first-order Taylor
>   expansion incurs a per-step error of $O(\Delta t)$ in the stochastic regime
>   (the $O(\sqrt{\Delta t})$ Brownian increment squared in the second-order term)
>   and $O(\Delta t^2)$ in the deterministic regime. A uniform horizon-wide bound
>   is highly specific to the drift, dimension, and application, and is beyond our
>   scope. Per-step linearization error is reported in Table 1, and we observed
>   zero final-sample violations across all experiments.
>
> - The assumption that $\mathcal{C}$ is closed is standard in constrained
>   optimization and control and nearly always holds in practice. We have removed
>   the boundedness assumption, which is not needed for the results.
>
> - This follows from Theorem 4.3 and is now clarified in the text. When $g=0$ the
>   process is deterministic and the objective $\|u\|^2/g(t)^2$ degenerates; the
>   KL functional (13) is no longer relevant and the natural analogue is the
>   integrated control energy $\int \|u\|^2 dt$. Minimizing $\|u\|^2$ pointwise is
>   then the minimal-$L_2$ drift perturbation keeping the trajectory in the tube,
>   i.e., the smallest deviation from the learned velocity field consistent with
>   safety. We have added a sentence to Section 4 making this explicit.

---

### Review · Reviewer_2bz9 · 2026-04-23

**Summary Of Contributions:**

The paper addresses how to constrain the diffusion process in generative models so that the generated samples lie exactly within a subset defined by analytical constraints. The proposed procedure can be summarized as follows: a time-dependent threshold is introduced along the diffusion process, such that at the beginning of the denoising phase the constraint is almost ineffective, and it gradually becomes stricter, ultimately enforcing the desired constraint at the end of the process. The authors illustrate this idea through the concept of a shrinking tube that progressively guides the diffusion toward the target region.
They derive the theoretical framework from the Control Barrier Functions approach in control theory and show that restricting to minimal-energy controls induces a bound on the KL divergence between the distributions obtained from the constrained and unconstrained diffusion processes.
Finally, the authors validate their framework across several settings, ranging from dynamical systems to image generation and action policies, demonstrating the effectiveness of their approach.

**Audience:**

Yes

**Audience Explanation:**

The topic of guided diffusion is fundamental nowadays, being related to generation of images, text and complex data with semantic structure. Even if still not applicable to latent diffusion as highlighted by the authors, the hope is that this mathematical foundation could be extended to that case; in conclusion, the paper represents a contribution that merits to be presented in my opinion to the community.

**Broader Impact Concerns:**

Since this could be potential applied in future to latent diffusion, I would spend a paragraph on concerns of misuse.

**Claims And Evidence:**

Yes

**Claims Explanation:**

I would like to emphasize that I found the paper very clear. In particular, I appreciated the effort devoted to providing intuition behind the mathematical results, as well as their clear manifestation in the experimental section. I did not identify any major weaknesses in the paper, and I will therefore suggest only minor corrections in the dedicated section below.
I also appreciated the space devoted to the Limitations section, where the authors clearly explain the rationale behind their methodological choices, including the challenges encountered and how they were addressed.

**Requested Changes:**

Here are some minor concerns that I would appreciate being addressed in the final version:

* Computational cost: What is the computational overhead of your approach compared to unconstrained diffusion and to state-of-the-art methods for safe generation? A comparative experiment in this direction would be very useful.
* Type of constraint: Why did you not report experiments with different types of constraints? I would expect that varying the constraint would affect the shrinking rate of the tube, which could be either beneficial or detrimental. It would be helpful to provide some intuition on this aspect.
* Non-convex sets: Could you present a (simple) experiment involving a non-convex constraint set? In particular, it would be interesting to observe what happens when convexity is lost during the generation process.
* Figure 2: In panel (b), you show that the control appears to act mainly in the initial steps of denoising. Could you compare this with a setting in which the constraint is switched off after t = 0.6 in the denoising process? It would be interesting to understand whether, in order to ensure that the final sample lies in the correct set, the constraint needs to be enforced only at the beginning of diffusion.
* Are there cases in which you can quantify how big is the error you make due to the linear approximation in the solution of the constrained problem?

---

> ### Author Response · Authors · 2026-05-29
> **Expanded empirical study: computational cost, non-convexity, and constriction schedules**
>
> - We now report computational overhead systematically. The per-step QP adds approximately 33 ms ($\approx$13\% over vanilla sampling) for image generation (Table 1) and increases inference from 47 ms to 63 ms ($\approx$34\%) for the robotics task (Table 2), which remains well within the real-time budget of the control loop.
>
> - We have added an explicitly non-convex experiment in Appendix A.2: the safe set is a union of two disjoint reference-window basins. The framework selects and renders one basin based solely on the initial noise, with no manual choice, and different seeds reach different basins, confirming that both components of the non-convex set are individually reachable.
>
> - We have added the experiment of cutting off the safe controller by our framework in Appendix A.3 (Figure 9). We release the controller at several cutoff times. Disabling control late ($t_{\text{off}}  = 0.05$) is indistinguishable from always-on, since $u^\ast \approx 0$ in the low-noise regime, but releasing during the high-noise phase ($t_{\text{off}} = 0.6, 0.9$) causes the constraint to fail as the model can still make updates to the sample. However, the process of coarse-to-fine constraint enforcement ensures that semantic information gets retained, although exact constraint satisfaction fails.
>
> - We report the maximum per-step linearization error directly in Table 1 (column $\max_k$ lin.err). We have also clarified its role in Section 4.3: the first-order Taylor expansion incurs a per-step error that is $O(\Delta t)$ in the stochastic regime (because the Brownian increment is $O(\sqrt{\Delta t})$, squared in the second-order term) and $O(\Delta t^2)$ in the deterministic regime. The class-$K$ margin absorbs this remainder under an explicit sufficient condition, yielding zero violations at the final sample across all experiments.
>
> - The three constriction schedules in Section 4.1 (linear, exponential, polynomial) differ in *when* along the sampling horizon the control effort is concentrated. Linear distributes intervention evenly, exponential ($\lambda > 1$) front-loads it to the high-noise regime where intervention is distributionally cheap (per Theorem 4.3), polynomial ($p > 1$) back-loads it to the low-noise regime. Section 5.2.3 (Table 1) compares all four schedules on the location/content constraint and reports both the maximum per-step linearization error and FID/KID/Vendi on the unconstrained region. The aggressive exponential $(\lambda = 3)$ minimizes linearization error, consistent with the $1/g(t)^2$ cost structure: concentrating constriction where the noise level is already large keeps the first-order Taylor approximation least strained. The back-loaded polynomial amplifies linearization error nearly four-fold. Generation quality on the unconstrained region is essentially independent of schedule.
>
> - We have now added a broader impact statement concerning the potential misuse of our exact guidance scheme.

---

### Review · Reviewer_fFKz · 2026-05-18

**Summary Of Contributions:**

This paper studies safe sampling for pre-trained flow-based generative models and proposes a guidance framework based on Control Barrier Functions (CBFs). The main idea is to define a constraining safety tube that is relaxed at the initial noise distribution and gradually tightens toward the target safe set during sampling, and to enforce this constraint by solving a minimum-norm QP at each step. The method is training-free, does not require architectural changes, and is evaluated on physics-consistent trajectory generation, constrained image generation, and smooth robot policy generation.

**Strengths:**

- The paper studies an important problem, namely how to enforce hard constraints in generative models with formal guarantees, which is relevant for safety-critical applications.

- The paper is well written and generally easy to follow.

- The proposed control-theoretic formulation is interesting, and the idea of using a constricting safety tube that matches the coarse-to-fine nature of diffusion sampling is well motivated.

- A practical advantage of the method is that it is training-free and can be applied to pre-trained models at sampling time.

**Weaknesses:**

- My main concern is with the experimental validation, which is currently limited. Although the paper shows that the constraints are satisfied in the presented settings, I would expect stronger qualitative and quantitative evidence to verify that the method works more broadly.

- In particular, the paper does not evaluate how the proposed guidance affects sample quality or diversity. Since the method modifies the sampling dynamics, it would be important to report metrics before and after guidance, for example quality scores such as FD or KID and diversity scores such as Vendi or RKE.

- The paper also lacks a clear analysis of the computational overhead. A more systematic comparison of memory and runtime costs across settings would make the practical efficiency of the method much clearer.

- One important limitation is that the method mainly operates in pixel space. I appreciate that the authors clearly discuss this limitation and even include preliminary latent diffusion experiments, but the paper itself shows that exact image-space constraint satisfaction does not transfer cleanly to latent diffusion because of decoder non-invertibility, which limits the current practical impact.

- The related work section can also be improved. The paper discusses projection-based and some reward-related methods, but it would benefit from a broader discussion of how this method relates to other guidance approaches in generative modeling, including reward guidance and other forms of controlled sampling.

**Additional Comments:**

Overall, I find this paper well motivated, technically interesting, and generally well written. I am positive about the contribution, especially because the paper tackles an important problem and provides a principled formulation with formal guarantees under clearly stated assumptions. My main concern is that the empirical section is not yet strong enough to fully support the practical impact of the method, and strengthening the quantitative evaluation would significantly improve the paper.

**Audience:**

Yes

**Audience Explanation:**

I believe this paper would be of interest to part of the TMLR audience, especially researchers working on generative models, safe and constrained sampling, robotics, and scientific machine learning. The problem is important, and the paper proposes a clean connection between control-theoretic safety tools and modern generative sampling.

Even with the current practical limitations, I think the contribution is meaningful. The paper offers a useful step toward bringing formal safety guarantees into sampling-based generative modeling, which is a topic that is becoming increasingly relevant.

**Broader Impact Concerns:**

I do not have major broader impact concerns beyond the standard issues associated with deploying generative models in safety-critical settings. Since the paper is explicitly motivated by safety, it may still be helpful for the authors to briefly discuss possible misuse scenarios and the limitations of guarantees when the barrier function is misspecified or unreliable.

**Claims And Evidence:**

Yes

**Claims Explanation:**

The theoretical part is one of the strongest aspects of the paper. The main safety guarantee is clearly stated, and the proof strategy for reverse invariance appears technically sound under the paper’s assumptions, especially the assumption that the controller has access to the realized noise at each sampling step. The KL argument is also reasonable, but it is important to note that the result supports greedy minimization of the instantaneous KL contribution, not global optimality over the full sampling horizon.

That said, the empirical evidence is less convincing than the theory. The paper demonstrates constraint satisfaction in the selected tasks, but the numerical evaluation is still limited and does not fully characterize the cost of enforcing safety. In particular, there is no quantitative analysis of whether the guidance harms generation quality or diversity, and there is also no comprehensive study of memory and runtime overhead.

Another point that deserves mention is the discrete-time implementation. The practical algorithm relies on a first-order Taylor approximation, and the paper explicitly notes that this yields only an approximate discrete-time guarantee without a formal step-size-dependent error bound. This does not invalidate the contribution, but it is an important caveat that should be reflected in the empirical evaluation.

**Requested Changes:**

**Important Changes:**

- The paper should include a stronger quantitative evaluation of the effect of guidance on the generated distribution. In particular, I would expect metrics for sample quality and diversity before and after guidance, such as FD/KID and diversity measures such as Vendi or RKE.

- The experimental section should include more qualitative results, especially for image generation. The current examples are limited, and additional results in the appendix would help verify that the method works consistently across different constraints and settings.

- The paper should provide a clearer analysis of computational overhead, including runtime and memory cost, not only for the robotics setting but more systematically across experiments.

**Other Changes:**

- The discussion of related work should be expanded to better position the method relative to other guidance approaches, including reward-guided and diversity/quality guidance methods.

- The paper would benefit from a clearer discussion of the assumptions behind the theoretical guarantees, especially the requirement of access to the realized noise and the fact that the discrete-time implementation is only approximately justified by the current analysis.

- The pixel-space limitation should be emphasized more clearly in the main discussion. I appreciate that the authors already acknowledge this point, and I still view the contribution positively, but this limitation does reduce the current practical scope of the method.

---

> ### Author Response · Authors · 2026-05-29
> **Quantitative distribution metrics, expanded experiments, and clarified assumptions**
>
> - We have added a quantitative study in Section 5.2.3 (Table 1). We report FID, KID, and Vendi scores for guided sampling against 500 unguided samples, computed on the unconstrained region so as to isolate the effect of guidance where the constraint is inactive. Across all constriction schedules, FID and Vendi remain close to the unguided reference (Vendi 6.62–6.66 vs. 7.35 unguided), indicating that hard guidance preserves generation quality and diversity in the unconstrained region. The residual diversity reduction is attributable to the shared forced content rather than degradation of the model.
>
> - We have expanded the appendix with several new sets of qualitative results: alternate reference images and placements (Appendix A.1), a non-convex disjunctive constraint across multiple seeds (Appendix A.2), a cutoff-steering ablation across five release times (Appendix A.3). Together these verify consistent behavior across different constraints and settings.
>
> - Computational overhead across experiments. We have made the overhead reporting systematic. The per-step QP overhead is approximately 33 ms ($\approx$13\% over vanilla) for image generation (Table 1) and increases inference from 47 ms to 63 ms ($\approx$34\%) for the robotics task (Table 2). Because the per-pixel barriers decompose into independent low-dimensional QPs (Section 5.2), memory overhead over the underlying model is negligible. Inference cost remains dominated by the model's forward passes.
>
> ## Other changes
>
> - We have expanded Section 2 to position our method relative to reward-guided and value-based guidance, and against the concurrent control-theoretic approach HardFlow. We note that reward-based methods grapple with a quality–diversity tradeoff, which directly motivates our quantitative study in Section 5.2.3.
>
> - We have clarified both assumptions the reviewer raises. The realized-noise requirement is now stated precisely in Section 3.1 (the observe-then-act structure of the simulated sampler) and reaffirmed in Section 4 and Remark 1: because sampling is simulated, each realized noise increment is observed before the control is synthesized, which is what permits the per-realization discrete-time guarantee. We have also restructured the safety guarantee so that the rigorous result (Theorem 4.2) is stated at the discrete-time level, with the continuous-time result repositioned as a formal proposition that motivates the design (Proposition 4.1). The discrete-time linearization error is analyzed in Section 4.3 in both stochastic and deterministic regimes, and we have added an explicit margin condition under which the linearized constraint implies the exact discrete constraint.
>
> -  We have emphasized this point in the main text following
> Proposition 4.1, and made the scope of the guarantee precise. Our guarantee holds exactly in all experiments where the constraint
> is imposed directly on the sampled variable. VAE decoders used in latent
> diffusion are not invertible maps between the sample space and the semantic space, so the latent-space certificate transfers to
> the decoded image up to the decoder's reconstruction error. We treat this case
> in Section 6 and demonstrate it empirically in Appendix A.4.